# Wiring variations that enable and constrain neural computation in a sensory microcircuit

William F Tobin, Rachel I Wilson*, Wei-Chung Allen Lee*[†]

Department of Neurobiology, Harvard Medical School, Boston, United States

**Abstract** Neural network function can be shaped by varying the strength of synaptic connections. One way to achieve this is to vary connection structure. To investigate how structural variation among synaptic connections might affect neural computation, we examined primary afferent connections in the *Drosophila* olfactory system. We used large-scale serial section electron microscopy to reconstruct all the olfactory receptor neuron (ORN) axons that target a left-right pair of glomeruli, as well as all the projection neurons (PNs) postsynaptic to these ORNs. We found three variations in ORN→PN connectivity. First, we found a systematic co-variation in synapse number and PN dendrite size, suggesting total synaptic conductance is tuned to postsynaptic excitability. Second, we discovered that PNs receive more synapses from ipsilateral than contralateral ORNs, providing a structural basis for odor lateralization behavior. Finally, we found evidence of imprecision in ORN→PN connections that can diminish network performance.

*For correspondence:
rachel_wilson@hms.harvard.edu
(RIW); wei-chung_lee@hms.
harvard.edu (W-CAL)

Present address: [†]F.M. Kirby
Neurobiology Center, Boston
Children's Hospital, Boston,
United States

Competing interests: The
authors declare that no
competing interests exist.

Reviewing editor: Liqun Luo,
Howard Hughes Medical
Institute, Stanford University,
United States

## Introduction

The wiring of a neural network is a key determinant of its function. In principle, each synaptic connection might have an optimal strength, as dictated by activity patterns passing through the synapses comprising that connection, as well as the computations that the connection ought to support. The strength of each connection will depend on many structural features, such as the number of presynaptic neurotransmitter release sites it contains, or the size of the postsynaptic dendrite.

For these reasons, it is important to understand how variations in the structure of synaptic connections will affect the computational properties of a network. The ability to create systematic variations in synapse structure ought to expand the potential computational performance of a network. Conversely, the performance of a network could be diminished if the structure of its synaptic connections is not specified precisely.

Even closely related synaptic connections can exhibit variations. For example, there can be anatomical differences in the connections made by an identifiable neuron in isogenic individuals (*Goodman, 1978*; *Ward et al., 1975*). Within an individual brain, there can be differences in connectivity across multiple instances of the same network module (*Takemura et al., 2015*). There can be anatomical differences in the connections made by right and left copies of an identifiable neuron in the same individual (*Lu et al., 2009*; *Ryan et al., 2016*). Finally, there can even be anatomical differences between the multiple synaptic contacts linking a presynaptic neuron with a single postsynaptic neuron (*Bartol et al., 2015*). Some of this observed variation is likely stochastic ('noise'), reflecting the limited precision of synapse specification. Some variation may be 'signal' – e.g., variation in synapse structure due to homeostatic compensation or plasticity. In general, it has often been difficult to discriminate between 'signal' and 'noise' (in this sense) in the analysis of connectomics data. It has also often been difficult to quantify how neural computations could be affected by observed variations in synapse structure.

In this study, we illustrate a general approach to solving these problems. We use serial section electron microscopy (EM) to reconstruct all the copies of a single cell type in one individual brain, along with all the feedforward excitatory synapses that these cells receive. We then combine these EM data with electrophysiological data to create constrained compartmental models of the reconstructed cells that allows us to quantify the functional implications of the wiring variations in this network. We are able to directly translate anatomical differences between synaptic connections (the number of synapses per connection, as well as the location of those synapses on the dendritic tree) into physiological differences (variations in synaptic voltage). This tells us which anatomical variations are likely functionally most important, how different anatomical parameters can interact (and potentially compensate for each other), and how anatomical variations can affect the amount of information transmitted across the synapse.

As a test-bed for this approach, we focus on a well-characterized microcircuit – a single olfactory glomerulus in the adult *Drosophila* antennal lobe. Each of the 50 glomeruli in this brain region receives feedforward excitation from a dedicated population of olfactory receptor neurons (ORNs); signals from each glomerulus are then relayed to higher brain regions by a dedicated population of projection neurons (PNs) (*Su et al., 2009*). All the ORNs presynaptic to each glomerulus express the same odorant receptor, and so have similar odor tuning (*Vosshall et al., 2000*). Each PN receives ORN input from all the ORNs whose axons enter its cognate glomerulus (*Kazama and Wilson, 2009*). ORN→PN connections are a good test-bed for studying wiring variations because there are so many connections per glomerulus, each with the same ostensible function. Moreover, the physiological properties of these connections have been extensively studied in vivo (*Gaudry et al., 2013*; *Gouwens and Wilson, 2009*; *Kazama and Wilson, 2008*, *2009*; *Olsen and Wilson, 2008*; *Root et al., 2008*).

This study has three parts. In the first part, we show that the number of synapses per ORN→PN connection matches the size of the PN dendrite, and this matching precisely compensates for size-dependent differences in dendritic filtering properties. In the second part of the study, we show that there is a systematic variation in the structure of ORN connections onto ipsi- versus contralateral PNs, and this difference is precise enough to allow the brain to lateralize odor stimuli to the left or right side of the head. In the third part of the study, we demonstrate there is some degree of unsystematic structural variation in ORN→PN synaptic connections ('noise'), and we show how this imprecision can constrain a PN's ability to count ORN spikes.

Taken together, our results imply that certain structural parameters of these synaptic connections are systematically varied to enable specific neural computations, but the limited precision of synaptic structural control also represents a fundamental constraint on network performance. More generally, our study illustrates a principled approach to partitioning 'signal' and 'noise' in connectomics data – i.e., systematic and unsystematic variations in wiring – with the goal of linking these variations to network function.

## Results

We used large-scale serial section transmission EM (*Bock et al., 2011*; *Lee et al., 2016*) to collect a volumetric data set comprising the anterior portion of one adult female *Drosophila* brain (*Figure 1A and B*, and *Video 1*). The series consisted of 1917 thin (<50 nm) sections cut in the frontal plane. Each section was acquired at 4 × 4 nm$^2$/pixel, amounting to 4 million camera images and 50 TB of raw data. Sections were imaged to capture both sides of the brain (up to 400 × 750 μm$^2$), including both right and left copies of the antennal lobe.

### Cells comprising the glomerular micro-circuit

We selected glomerulus DM6 for reconstruction because it is easily identifiable and because there is a large amount of published data on the physiology of DM6 PNs (*Gaudry et al., 2013*; *Kazama and Wilson, 2008*, *2009*). Three experts independently identified glomerulus DM6 in the EM data set by visual inspection based on published light-level maps (*Couto et al., 2005*; *Grabe et al., 2015*). We then manually reconstructed and validated all the ORNs and PNs targeting glomerulus DM6 (see Materials and methods; *Figure 1C,D*, *Figure 1—figure supplement 1*, and *Video 2*). We confirmed ORN identity by tracing each axon to the antennal nerve. We confirmed that all reconstructed ORNs innervated DM6 and no other glomeruli. In total, 53 ORNs innervated DM6, with 27 axons coming

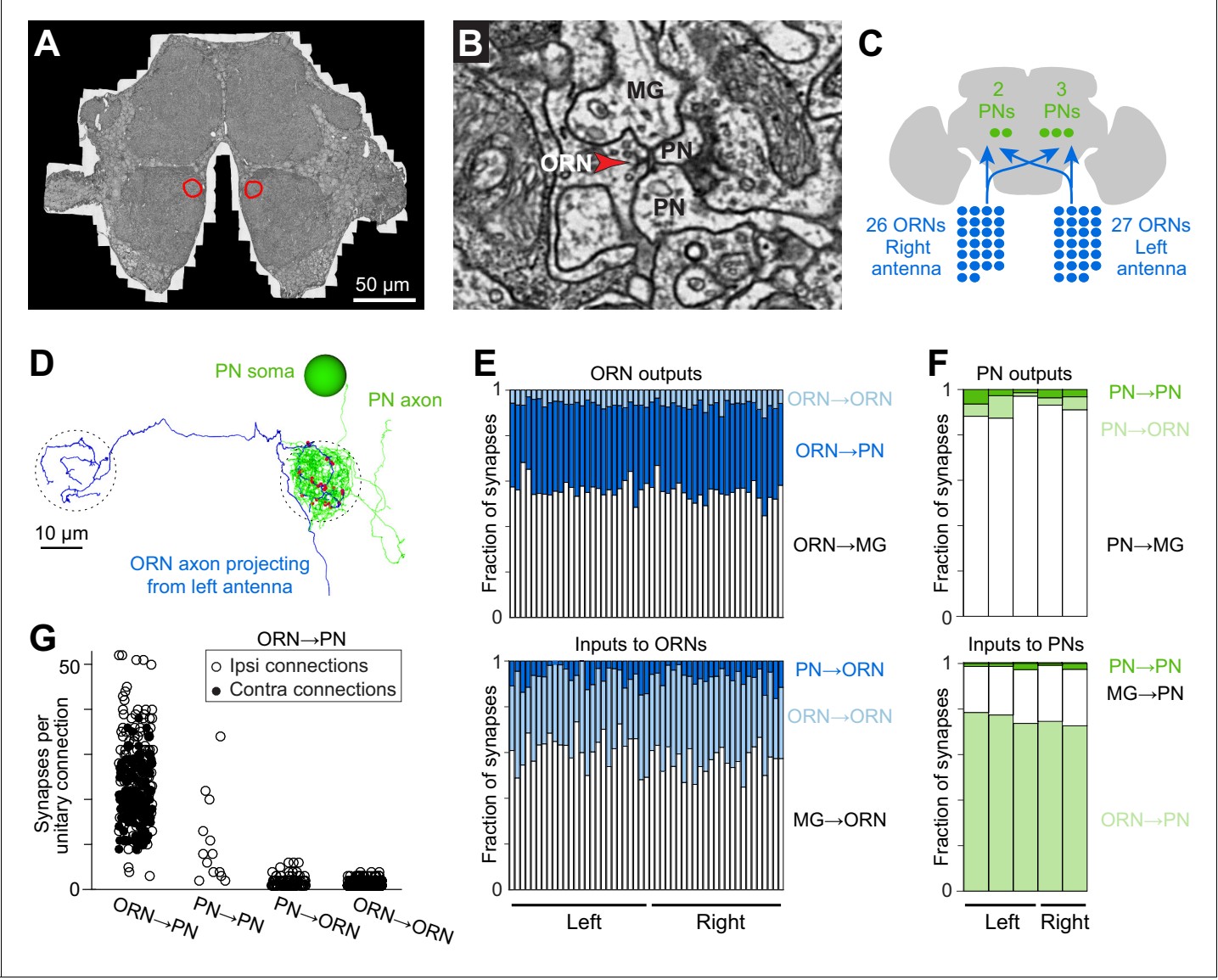

**Figure 1.** Cells and connections comprising the glomerular micro-circuit. (**A**) Electron micrograph of a frontal section (76,000 × 58,000 pixels) from the anterior portion of the brain. Glomerulus DM6 is outlined. (**B**) Zoomed-in view of a synapse. Red arrowhead demarcates a presynaptic specialization (T-bar). The long edge of the image measures 2 μm. (**C**) Schematic of reconstructed cells. All ORNs and PNs within glomerulus DM6 were fully reconstructed. Not shown in this schematic are multi-glomerular neurons (cells that interconnect different glomeruli), which were not fully reconstructed. (**D**) 3-D rendering of a single ORN→PN cell pair viewed frontally. The ORN axon (blue) makes multiple synapses (red) onto the PN dendrite (green). The approximate region occupied by the DM6 glomerulus is circled with dashed lines. The PN cell body is represented as a sphere for display purposes. (**E**) Top: synapses made by ORNs, expressed as a fraction of each ORN's total pool of output synapses. Bottom: synapses received by ORNs, expressed as a fraction of each ORN's total pool of input synapses. 'MG' denotes multiglomerular neurons. (**F**) Synapses made by PNs and received by PNs, normalized in the same way. (**G**) Number of synapses between connected pairs of cells, sorted by connection type. Each symbol represents a unitary connection.

The following source data and figure supplements are available for figure 1:

**Figure supplement 1.** Anatomical gallery: ORNs, PNs, and connected pairs.

**Figure supplement 2.** Variations in the number of PNs in glomerulus DM6.

**Figure supplement 3.** Images of representative synapses connecting different cell types.

**Figure supplement 4.** Connectivity matrix of ORNs and PNs in glomerulus DM6.

*Figure 1 continued on next page*

*Figure 1 continued*

**Figure supplement 4—source data 1.** Matrix of ORN-PN connectivity.

**Figure supplement 5.** Multiplicity of postsynaptic profiles.

**Figure supplement 6.** T-bar volume and postsynaptic contact area.

from the left antenna and 26 from the right. Like most *Drosophila* ORNs, these cells project bilaterally (*Vosshall et al., 2000*; *Couto et al., 2005*). However, we found that one right ORN and one left ORN projected only ipsilaterally. Therefore, each glomerulus contained the axons of 52 ORNs. We also reconstructed the dendrite, soma, and primary neurite of each DM6 PN, and confirmed that these cells sent their axons into the medial antennal lobe tract. The morphologies of all reconstructed cells are shown in *Figure 1—figure supplement 1*.

Interestingly, we found three 'sister' PNs on the left side the brain and two on the right. This is not an unusual occurrence: we found that the number of DM6 PNs is often different on the two sides of the brain, although with no consistent trend for left PNs to outnumber right PNs, or vice versa (*Figure 1—figure supplement 2*). Thus, variations in neuron number appear to be general phenomenon. This is consistent with a previous finding that certain identifiable cells are sometimes missing from a particular column in the *Drosophila* optic lobe (*Takemura et al., 2015*). In a later section, we will examine in detail the issue of cell number variation and its network correlates.

Profiles not identified as ORNs or PNs are referred to here as multiglomerular neurons. Most of these profiles are probably inhibitory local neurons, because this cell type is the most numerous and broadly-arborizing of the multiglomerular cell types in the antennal lobe (*Chou et al., 2010*; *Lin et al., 2012*). In addition, some of these profiles may be multiglomerular projection neurons, or neurons that project into the antennal lobe from other regions.

To estimate our reconstruction error, we quantified the percentage of cellular profiles presynaptic to PNs that were 'orphans', i.e. fragments of reconstructed neurites that could not be connected to any neuron. Overall, the percentage of orphans was relatively small (6.8 ± 1.5%, mean ± SEM). This value was lower for the left glomerulus (4.4 ± 0.8%) than for the right glomerulus (10.4 ± 0.4%). Another study using a similar reconstruction strategy demonstrated that false continuations between fragments are easily detected and corrected during independent review, and are therefore very rare in the final reconstruction (*Schneider-Mizell et al., 2016*).

## Relative abundance of connection types

In addition to reconstructing the morphology of every ORN and PN in glomerulus DM6, we also identified all their pre- and postsynaptic specializations. Images of representative synapses are shown in *Figure 1—figure supplement 3*. We confirmed the existence of every connection type previously reported in the antennal lobe of the adult fly (*Rybak et al., 2016*). The full matrix of ORN-PN connectivity is shown in *Figure 1—figure supplement 4* and *Figure 1—figure supplement 4—source data 1*.

Beginning with ORN output synapses, we found that ORNs delivered about one-third of their synaptic output to PNs (*Figure 1E*). Most of the remainder was delivered to multiglomerular neurons (*Figure 1E*). Because most multiglomerular neurons are likely inhibitory local neurons, this pattern emphasizes the important

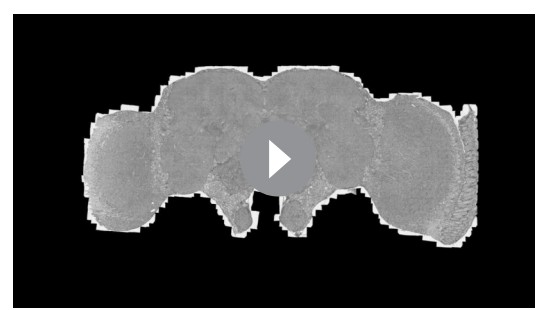

**Video 1.** EM volume of the anterior fly brain. The video shows a fly-through (posterior to anterior) of the aligned EM series. Please see Data Availability for directions to the publicly accessible high-resolution aligned data set.

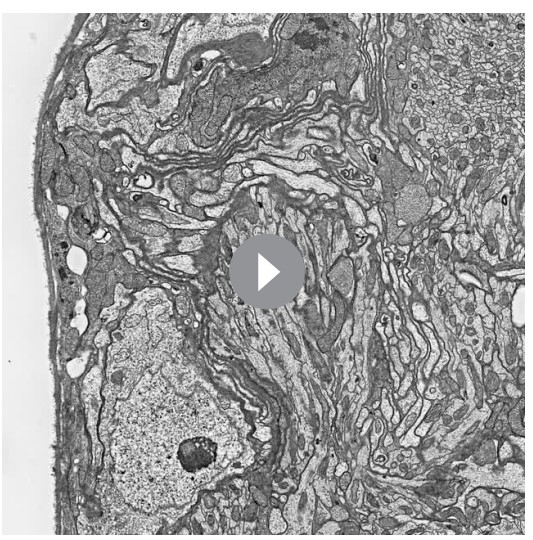

**Video 2.** Serial EM sections through left glomerulus DM6. The video shows a fly-through (anterior to posterior) of a cropped (16.4 µm × 16.4 µm) volume traversing 329 of the aligned EM sections containing glomerulus DM6 neuropil in the left hemisphere of the brain.

role of feedforward inhibition in the antennal lobe circuit.

ORN axon terminals are not just presynaptic elements – they are also postsynaptic to other neurons (*Berck et al., 2016*; *Rybak et al., 2016*). We found that DM6 ORN axons received most of their synaptic input from multiglomerular neurons (*Figure 1E*). This is consistent with physiological studies showing that inhibitory local neurons exert potent control of neurotransmitter release from ORN axon terminals (*Olsen and Wilson, 2008*; *Root et al., 2008*). About a third of synapses onto ORN axon terminals originated from PNs and ORNs. It is not known how PN and ORN inputs might affect ORN axon terminals; in principle, they might either facilitate or suppress neurotransmitter release.

Turning to the PN's perspective, we found that PNs received most of their input (~75%) from ORNs, as expected (*Figure 1F*). PNs also received a sizeable input (~20%) from multiglomerular neurons. This is also expected, as paired electrophysiological recordings have demonstrated the existence of functional inhibitory synapses from local neurons onto PN dendrites

(*Liu and Wilson, 2013*; *Yaksi and Wilson, 2010*).

PN dendrites are known to contain presynaptic elements as well as postsynaptic elements (*Ng et al., 2002*). In this regard, PNs are analogous to olfactory bulb mitral/tufted cells, which also release neurotransmitter from their dendrites onto other cells in the same glomerulus (*Urban and Sakmann, 2002*). Within the DM6 glomerulus, we observed that PNs devoted almost all of their output synapses (~90%) to multiglomerular neurons (*Figure 1F*). PNs devoted a small part of their output to ORN axon terminals. Finally, every PN also made connections with all the other PNs in the same glomerulus, consistent with electrophysiological evidence for reciprocal synaptic interactions between sister DM6 PNs (*Kazama and Wilson, 2009*).

## The structure of excitatory connections

We will use the term 'unitary connection' to refer to all the synapses between one presynaptic cell and one postsynaptic cell. Most of the unitary connections we detected were composed of multiple synapses – i.e., multiple contacts at distinct locations, each with its own presynaptic specialization. This is illustrated by the ORN-PN pairs shown in *Figure 1D* (*Figure 1—figure supplement 1*).

On average, unitary ORN→PN connections were composed of about 23 synapses (*Figure 1G*). In principle, a connection composed of so many individual synapses should be both strong and reliable. Indeed, physiological studies have shown that each ORN axon makes a strong and reliable excitatory connection onto every PN in its target glomerulus (*Gaudry et al., 2013*; *Kazama and Wilson, 2008*, *2009*). An isolated ORN spike typically depolarizes a PN by about 5 mV, and the size of this unitary excitatory postsynaptic potential (uEPSP) is highly reliable from trial to trial. The average number of synapses that we detected per unitary ORN→PN connection is similar to that predicted by quantal analysis in whole-cell recordings (*Kazama and Wilson, 2008*) and also by light microscopy methods (*Mosca and Luo, 2014*).

Compared to ORN→PN connections, the other connections involving ORNs and PNs were weaker, as judged by the number of synapses they contained. Unitary PN→PN connections were composed of about 11 synapses, on average (*Figure 1G*). ORN→ORN connections and PN→ORN connections typically consisted of just one or two synapses. However, the influence of ORN→ORN connections in particular is likely to be non-negligible, because there are so many ORNs in total, and so ORNs collectively provide a major input to other ORNs (*Figure 1E*).

For the remainder of this study, we will focus on ORN→PN synapses and connections. It should be noted that each ORN presynaptic element (i.e., each 'T-bar') is presynaptic to multiple postsynaptic profiles, often one PN profile and one or two non-PN profiles (*Figure 1—figure supplement 5*). This sort of 'sharing' among postsynaptic elements is typical of *Drosophila* CNS synapses (*Prokop and Meinertzhagen, 2006*). The size of a T-bar was strongly correlated with the number of postsynaptic profiles; it was also significantly (though weakly) correlated with the surface area of the postsynaptic contacts (*Figure 1—figure supplement 6*). These observations suggest that there are mechanisms that coordinate the overall scale of a T-bar and its associated postsynaptic structures, producing concerted pre- and postsynaptic growth. On a synapse-by-synapse basis, these synapse size variations are relatively large; however, because each ORN→PN connection comprises multiple synapses, and because the size variations across synapses tend to average out among the different synapses within each connection (*Figure 1—figure supplement 6*), the overall functional impact of these synapse size variations is probably relatively small.

## Compensatory variations in dendrite size and synapse number

Because glomerulus DM6 exists on both the right and left sides of the brain, our data set provided an opportunity to examine two copies of the same microcircuit that developed under the same genetic program with almost identical sensory experiences (assuming no systematic differences between ORN activity in the right and left antennae). In the brain where we performed our EM reconstructions, we found two DM6 PNs on the right side and three on the left (*Figure 2A*). This sort of right-left asymmetry is evidently common: in a larger set of brains, we found that the number of PNs in glomerulus DM6 varies between two and four, and PN counts are often different on the right and left (*Figure 1—figure supplement 2*). Related to the left-right asymmetry in PN number, we found a number of differences between the left and right versions of DM6 that provide insight into compensatory mechanisms that circuits may employ to contend with variations in cell numbers.

The first thing we noticed was that the total dendritic path length was larger for individual PNs on the right as compared to individual PNs on the left (*Figure 2A*), possibly because right PNs had fewer sister cells, and so their dendrites had more space to fill. Notably, the total number of ORN→PN synapses was similar in the left and right glomerulus (*Figure 2B*). However, because there were three left PNs but only two right PNs, the number of ORN synapses received by an individual PN was 53% higher on the right than on the left. Because PNs on the right also had larger dendrites, this resulted in a similar ORN→PN synapse density in left and right PNs (*Figure 2C*). Finally, the number of multiglomerular neuron synapses received by an individual PN was also higher on the right than on the left (*Figure 2D*), and individual PNs on the right formed more presynaptic contacts than did individual PNs on the left (*Figure 2E*).

From a PN's perspective, which of these factors has the largest effect on the strength of synaptic connections? Is it synapse number, synapse density, or total dendrite size? Connection strength should grow with synapse number, and therefore ORN→PN connections might be stronger on the right. However, connection strength should diminish as the overall size of a dendritic arbor increases (because large arbors have lower input resistance). To deduce how these factors interact, we used compartmental modeling, an established analytical approach for deducing how synaptic conductances are translated into postsynaptic voltages (*Rall, 1964*). We divided each reconstructed PN into large numbers of very short segments (14,510–23,502 compartments per PN). Compartments were represented as passive isopotential units connected by resistors (*Figure 3A*). We took the specific membrane resistance and capacitance from a previous electrophysiological study which measured these values directly (*Gouwens and Wilson, 2009*). To simulate synaptic events, we opened conductances in the compartments where postsynaptic sites were identified in our reconstructions, and we analyzed the resulting voltage changes as they propagated throughout the dendritic tree and to the cell body. This allows us to see how measured anatomical variations in these cells can interact to shape PN voltage responses. The time course of the synaptic conductance was taken from a previous study (*Gouwens and Wilson, 2009*), and the peak of the synaptic conductance was set to a value which generated postsynaptic potentials with a realistic size (~5 mV, taken from *Kazama and Wilson, 2008*). All synaptic conductance events had the same size and shape, as if a presynaptic spike always opened a fixed number of postsynaptic neurotransmitter receptors with fixed properties; this simplification is reasonable given that there are only small differences across connections in the overall scale of T-bars and postsynaptic contacts (*Figure 1—figure supplement 6*). We will use

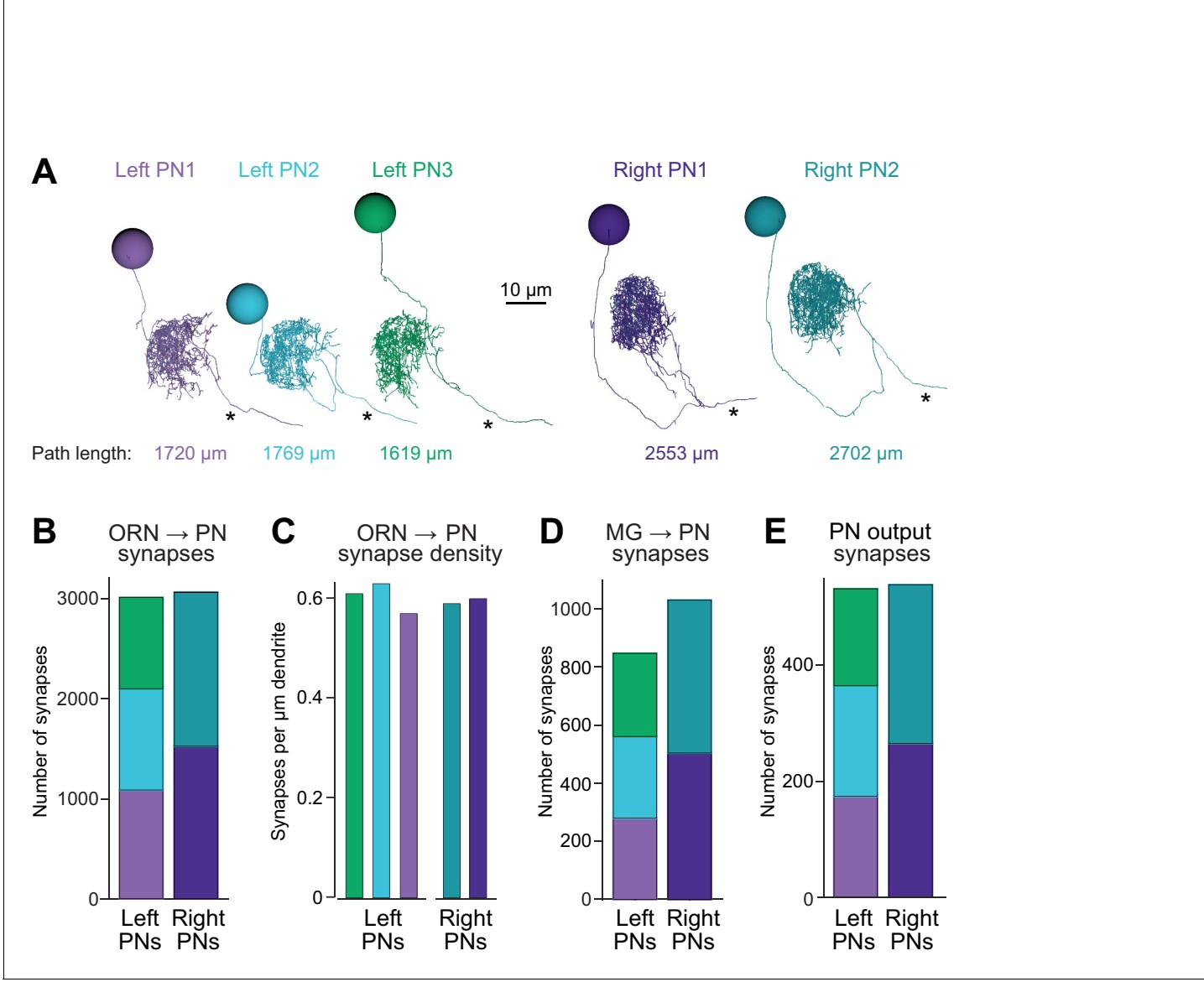

**Figure 2.** Connectivity compensates for a missing cell. (**A**) Skeletonized 3-D renderings of reconstructed PNs. Cells are viewed parasagittally from the left. For each cell, the total path length of all dendrite segments is indicated; note that PNs on the right side of the brain have longer path lengths. Axons are indicated with asterisks. (**B**) Synapses received by each PN from ORNs. (**C**) Number of ORN→PN synapses for each PN, normalized for the total path length of each PN's dendrites. (**D**) Synapses received by each PN from multi-glomerular (MG) neurons. (**E**) Output synapses made by each PN within the DM6 glomerulus.

our compartmental models as analytical tools for exploring the functional implications of 3-D ultra-structure data.

These models allowed us to determine if right and left PNs will respond differently to ORN spikes, given countervailing asymmetries in synapse number and dendritic size. For each of the 260 ORN→PN connections in our reconstruction, we used the models to infer the corresponding unitary excitatory postsynaptic potential (uEPSP) in the PN cell body (*Figure 3B*). Surprisingly, the simulated somatic uEPSP amplitudes were almost identical in right and left PNs (*Figure 3C*). In other words, the average strength of unitary connections was the same on the right and left. This is notable because of the substantial difference in the number of synapses per ORN→PN connection on the right and the left (*Figure 2B*).

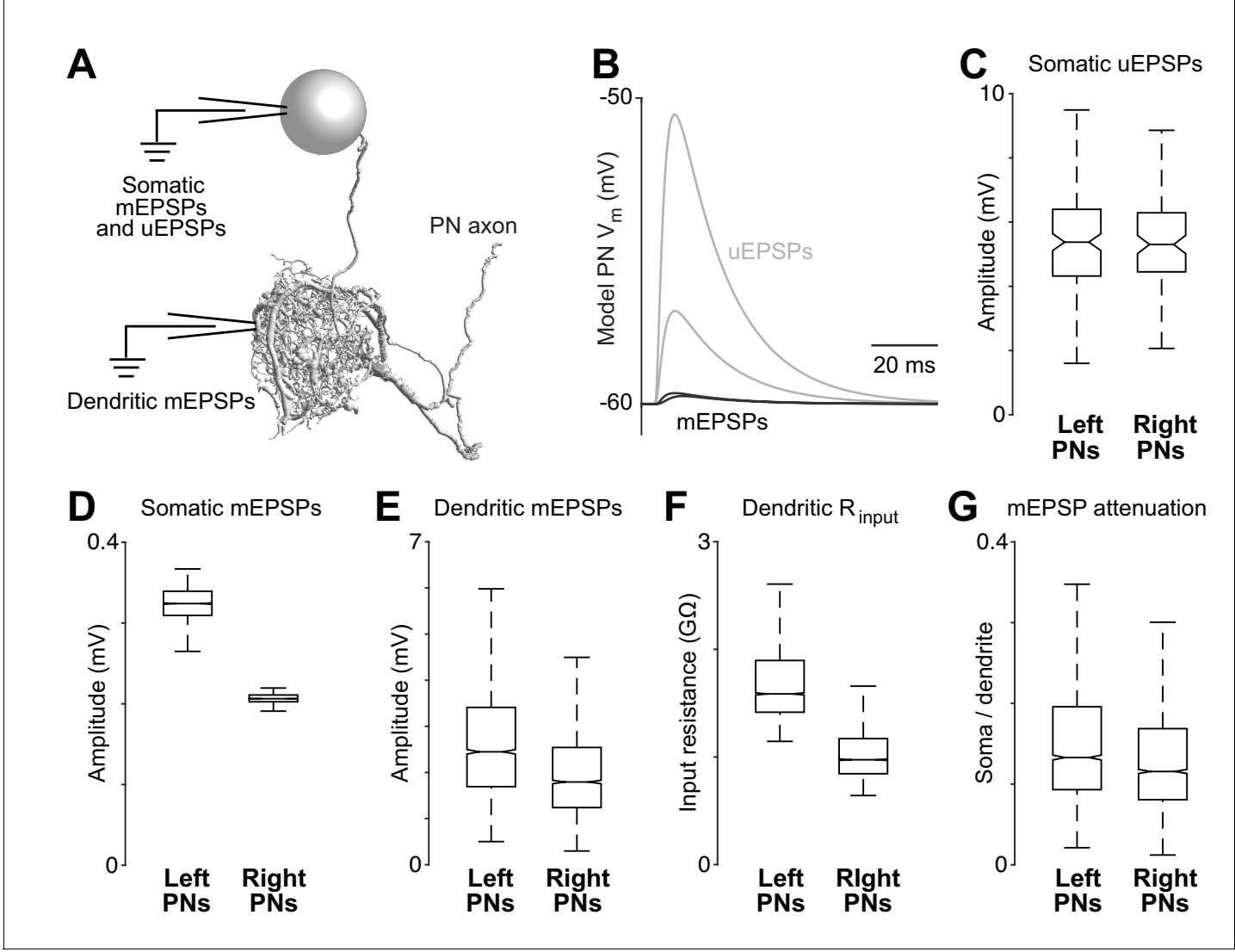

**Figure 3.** Dendritic arbor size equalizes average unitary responses. (A) Volumetric 3-D rendering of an EM reconstructed PN dendrite. (The cell body is represented as a sphere for display purposes.) Compartmental models fit to ultrastructural and electrophysiological data were used to simulate the voltage responses of each PN to synaptic input from ORNs. Synaptic conductances were simulated in the PN dendrite, and voltage responses were recorded in either the cell body or in the dendritic compartment where a given synapse was located. Each of the five reconstructed PNs was modeled in this way. (B) Example voltage responses recorded at the cell body of a model PN. A miniature EPSP (mEPSP) is the response to a quantum of neurotransmitter. A unitary EPSP (uEPSP) is the response to one spike in a single presynaptic axon, i.e. the combined effect of all the mEPSPs generated by that axon. Shown here are the largest and smallest mEPSPs and uEPSPs in this PN. In the model, a spike always produces the same conductance at all synapses, and so variations in mEPSP amplitude must be due to variations in the position of synapses on the dendrite. (C) There is no left-right difference in the amplitude of uEPSPs, measured at the cell body of each PN (computed across unitary ORN→PN connections, p>0.7, permutation test, *n* = 156 left and 104 right unitary connections). Here and elsewhere, box plots show median, 25th percentile and 75th percentile. Whiskers indicate 2.7 SDs (99.3% coverage of normally distributed data); for clarity, outliers beyond the whiskers are not displayed; notches indicate 95% confidence intervals. (D) There is a significant left-right difference in the amplitude of mEPSPs, measured at the cell body of the PN (computed across all ORN→PN synapses, p<0.0001, permutation test, *n* = 3013 left and 3066 right synapses). (E) There is a significant left-right difference in the amplitude of mEPSPs, measured at the site of each synapse in the dendrite (computed across all ORN→PN synapses, p<0.0001, permutation test). (F) There is a significant left-right difference in dendritic input resistance, measured across all model cables (computed across all PN cables, p<0.0001, permutation test, *n* = 5520 left and 6048 right cables). (G) There is a significant left-right difference in the attenuation of mEPSPs as they travel from the site of the synapse to the soma (ratio of somatic to dendritic amplitude computed across all ORN→PN synapses, p<0.0001, permutation test).

To better understand this result, we decomposed uEPSPs into their elemental components – i.e., miniature excitatory postsynaptic potentials (mEPSPs). When we measured mEPSPs at the soma, they were much smaller in right PNs than in left PNs (*Figure 3D*). The difference in mEPSP amplitude also appeared when we measured each mEPSP in the dendrite, at the site of the synapse itself (*Figure 3E*), reflecting a lower dendritic input resistance in right PNs (*Figure 3F*). Moreover, in right PNs, mEPSPs also decayed significantly more as they propagated from the dendrite to the cell body (*Figure 3G*), reflecting a longer average path length from synapse to cell body (*Figure 2A*). These differences highlight the important effect of the dendrite's overall morphology.

In summary, we find that right and left PNs have uniform average voltage responses to ORN spikes, in spite of their marked differences in dendrite size and synapse number. In principle, synapse number might be altered to compensate for dendrite size, or vice versa. Whatever the mechanism that creates this compensatory effect, the key point is that the measured differences in synapse number precisely counterbalance the measured differences in dendrite size.

## A basis for odor lateralization in the structure of synaptic connections

*Drosophila* can lateralize odors by comparing ORN spike trains arising from the right and left antennae. In response to a laterally asymmetric odor stimulus, flies will tend to turn toward the antenna that is stimulated more strongly (*Figure 4A*) (*Borst, 1983*; *Gaudry et al., 2013*). Because fly ORNs project bilaterally (*Figure 4B*), odor lateralization would be impossible unless there were an asymmetry between ORN connections in the ipsi- and contralateral antennal lobes. Electrophysiological recordings have shown that unitary ORN→PN connections are 30–40% stronger on the side of the brain ipsilateral to the ORN's cell body (*Gaudry et al., 2013*). We therefore asked if there is a structural basis for this ipsi/contra difference.

Indeed, we found that PNs received more input synapses from ipsilateral ORNs than from contralateral ORNs (*Figure 4C*). Accordingly, our compartmental models predicted that ipsilateral uEPSPs are 27% stronger than contralateral uEPSPs, on average (*Figure 4D*). There was no systematic ipsi-contra difference in the placement of synapses onto the PN's dendritic tree, as judged by the mean mEPSP amplitude at the PN soma (*Figure 4E*). There was also no significant difference in the total path length of ipsi- and contralateral ORN axons within the boundaries of glomerulus DM6 (*Figure 4—figure supplement 1*).

In short, we find that the difference in the electrophysiological properties of ipsi- and contralateral connections is largely explained by a single structural difference: PNs receive more synapses from ipsilateral than from contralateral ORNs. The ipsi-contra asymmetry is not perfectly precise at the level of single synaptic connections: some ipsi connections are weaker than contra connections onto the same PN (*Figure 1—figure supplement 4*). However, as we will show below, this difference is sufficiently precise to permit PNs to prefer ipsilateral stimuli over contralateral stimuli, provided that the stimuli are sufficiently strong. Thus, this finding represents a case where a particular behavior (odor lateralization) can be traced to a systematic variation in structure of connections. It also raises the interesting developmental problem of how ORN axons distinguish ipsilateral from contralateral glomeruli and adjust synapse number accordingly.

## Inequalities among olfactory receptor neurons in the same antenna

Individual neurons can improve the signal-to-noise ratio of their spike trains by pooling many inputs carrying a common signal but independent noise. The olfactory system is often cited as one of the clearest examples of this strategy. By pooling across many ORNs that express the same odorant receptor, a postsynaptic neuron should be able to dramatically improve the trial-to-trial reliability of its odor responses. If we assume that all sister ORNs are equally reliable, then the optimal strategy is to weight them equally. Alternatively, if some sister ORNs are more reliable than others, then the optimal strategy is to weight these inputs more heavily. We therefore examined the structure of ORN→PN connections for clues as to how ORNs are weighted.

We focused first on connections made by ipsilateral ORN axons, so as to remove the factor of ipsi-contra differences (*Figure 5A*). Even among ipsilateral ORN axons, we found substantial structural variation among the connections that they formed with PN dendrites. The main source of variation was the number of synapses per connection. This number varied over a wide range: for example, a typical PN received only 10 synapses from one ipsilateral ORN, but 36 synapses from

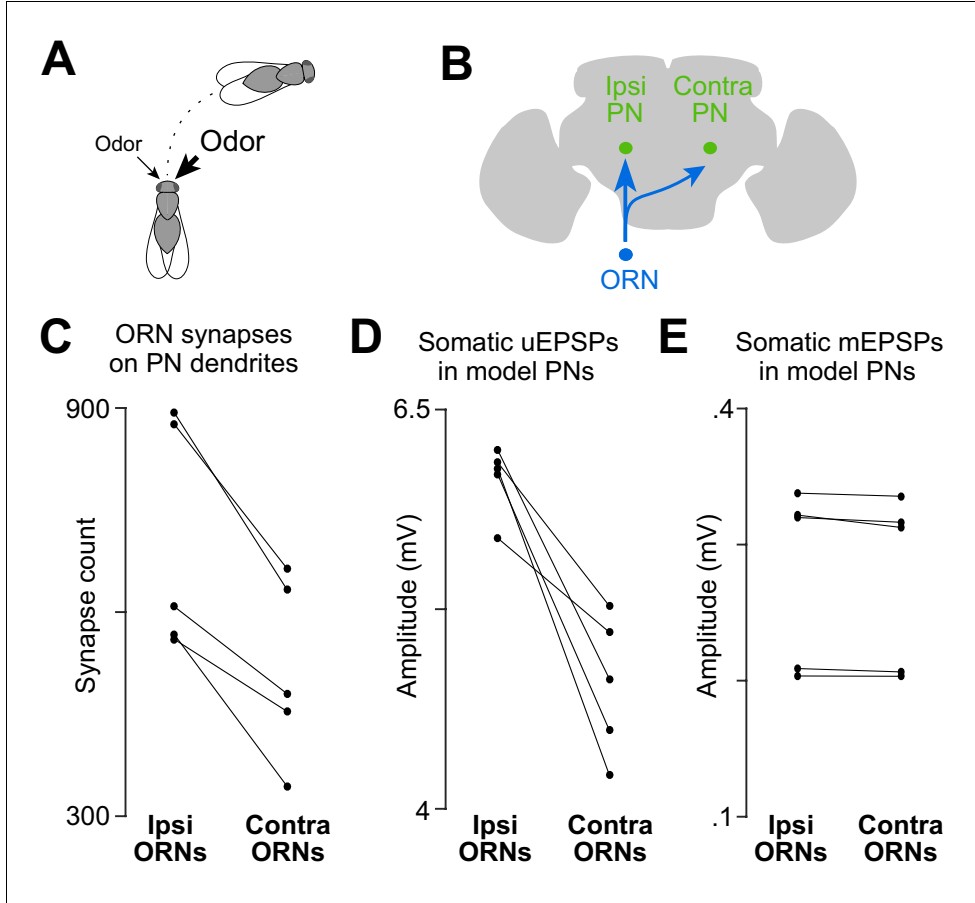

**Figure 4.** A basis for odor lateralization behavior in ORN wiring. (**A**) Flies turn toward lateralized odor stimuli, a behavioral response termed osmotropotaxis. (**B**) Schematic of an ORN axon projecting bilaterally. Note that ipsi and contra are defined relative to the location of the ORN cell body. Unitary EPSPs in PNs driven by ipsilateral ORNs are systematically larger than those driven by contralateral ORNs (*Gaudry et al., 2013*). Successful odor lateralization requires that ipsi- and contralateral connections are systematically different. (**C**) PNs receive significantly more synapses from ipsilateral ORNs than from contralateral ORNs. Each connected pair of points represents a PN (p=0.0032, paired-sample *t*-test, *n* = 5 PNs). These 5 PNs collectively have 133 ipsi and 132 contra connections. (**D**) There is a significant ipsi-contra difference in mean modeled uEPSP amplitudes. Each connected pair of points represents a PN, with values averaged across all the connections received by that PN (p=0.0059, paired-sample *t*-test, *n* = 5 PNs). (**E**) There is no ipsi-contra difference in modeled mEPSP amplitudes. Each connected pair of points represents a PN, with values averaged over all the synapses received by that PN (p>0.05, paired-sample *t*-test, *n* = 5 PNs). These 5 PNs collectively have 3504 ipsi and 2575 contra synapses).

The following figure supplement is available for figure 4:

**Figure supplement 1.** Comparing anatomical features of ipsi- and contralateral ORN axons.

---

another ipsilateral ORN (left PN1, *Figure 5B*). The number of synapses per connection was correlated with the physical proximity of ORN axons and PN dendrites (*Figure 5—figure supplement 1*), suggesting the number of synapses connecting each ORN-PN pair simply scales with the amount of axon-dendrite contact.

To normalize for the fact that different PNs receive different numbers of synapses, we divided the number of synapses per ORN→PN connection by each PN's total number of synapses received from ORNs (*Figure 5C*). This value expresses the contribution of each ORN to a PN pool of ORN synapses. Across all connections, the coefficient of variation (CV) in this value was 0.31, consistent with previous estimates based on optical methods (*Mosca and Luo, 2014*).

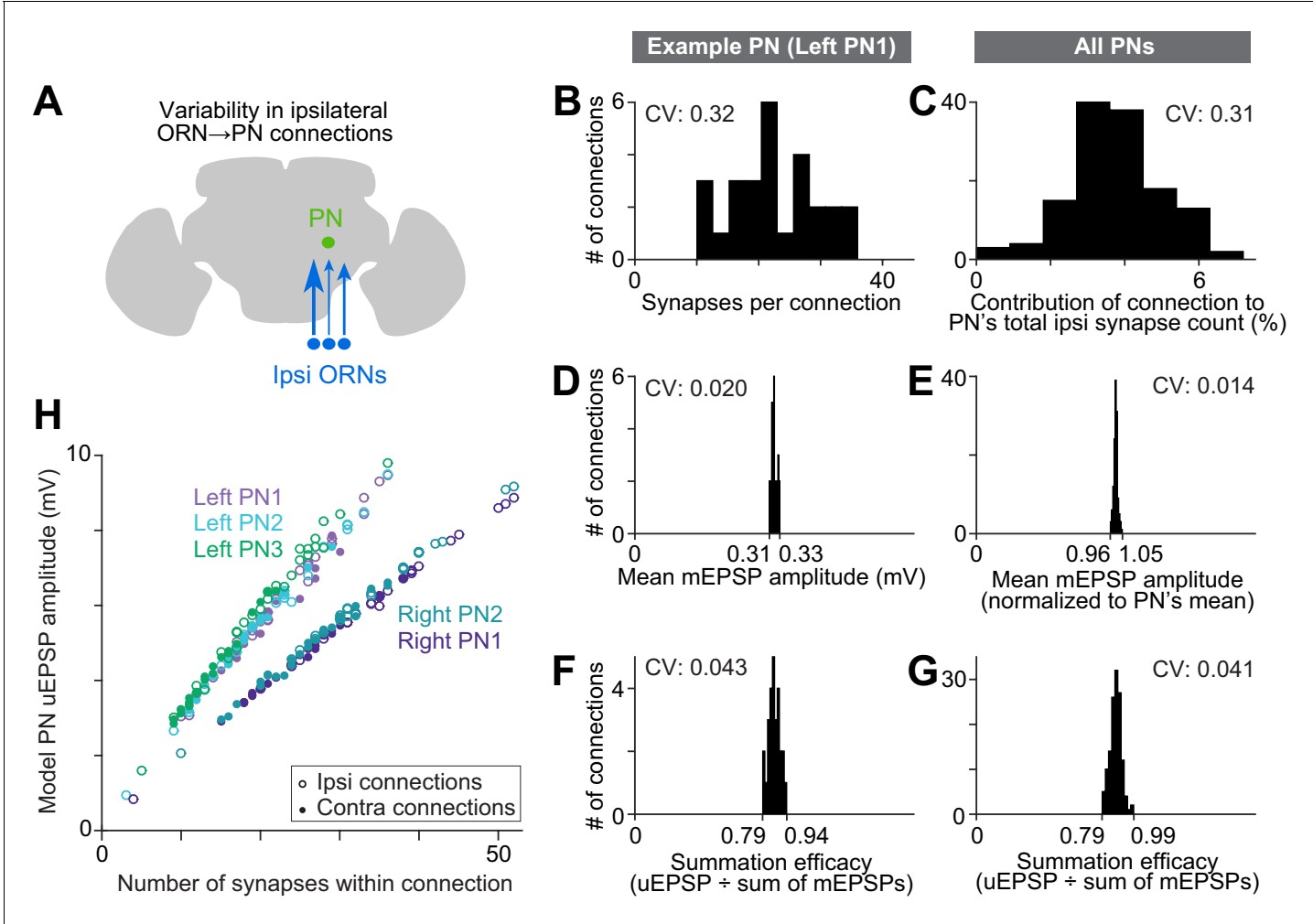

**Figure 5.** Wiring inequalities among sister ORNs. (A) Schematic of ORNs connected to a PN with different numbers of synapses. Arrow size represents the number of synapses each ORN forms on an ipsilateral PN dendrite. (B–G) Histograms showing variation among ipsilateral ORN→PN connections. The histograms are horizontally scaled so that the means of all distributions are aligned, in order to an enable a visual comparison of CVs. (B) Number of synapses that each ipsilateral ORN makes onto left PN1. (C) Analogous to (B) but pooled across all PNs. To enable pooling, we first normalize the number of synapses made by each ipsilateral ORN to the total number of ipsilateral ORN input synapses a PN receives. This yields the percentage contribution of each ORN to the ipsilateral synapse pool. The mean of this value is relatively consistent across the five PNs (0.037, 0.037, 0.037, 0.039, and 0.039), but there is a large variation within each PN. (D) Mean mEPSP amplitude for connections made by ipsilateral ORNs onto left PN1. At each unitary connection, the mean mEPSP amplitude is computed across all the synapses that contribute to that connection. This value is relatively consistent across unitary connections. (E) Analogous to (D) but pooled across all PNs. Each mEPSP value is normalized to the grand average for that PN. This value is consistent across all unitary connections, both within and across PNs. (F) Summation efficacy at connections made by ipsilateral ORNs onto left PN1. Summation efficacy is computed as the amplitude of the connection's uEPSP, divided by the linearly summed amplitudes of all the mEPSPs that comprise the connection. Again, this value is relatively consistent across unitary connections. (G) Same as (F) but pooled across all PNs. (H) Correlation between synapse number per connection and uEPSP amplitude. Each data point is a unitary connection ($n = 260$), with ipsilateral (unfilled) and contralateral (filled) connections indicated. For each PN, there is a strong and significant correlation (Pearson's $r$ ranges from 0.993 to 0.999; P ranges from $9.78 \times 10^{-48}$ to $1.34 \times 10^{-65}$ after Bonferroni-Holm correction for multiple comparisons, $m = 5$ tests).

The following figure supplement is available for figure 5:

**Figure supplement 1.** Axon-dendrite proximity correlates with the number of synapses per ORN→PN connection.

By comparison, there was little variability among connections in the mean amplitude of a simulated mEPSP at the soma of the compartmental models (*Figure 5D and E*). This is because each connection comprised many synapses, and there was little systematic variation across connections in the placement of the synapses on the dendrite. Summation efficacy was also relatively consistent across ORN→PN connections (*Figure 5F and G*). Summation efficacy is measured as the amplitude of the uEPSP, divided by the summed amplitudes of all the mEPSPs that contribute to that connection. Most connections had summation efficacies near 0.9, indicating weakly sublinear summation.

Why are average mEPSP amplitude and summation efficacy so consistent (*Figure 5D–G*)? We obtain a similar level of consistency if we randomly allocate synapses to ORN axons. Specifically, we allocate to each ORN axon the same number of synapses as before, but we draw synapses randomly (without replacement) from the pool of ORN→PN synapse locations. After shuffling, average mEPSP amplitude is still consistent across connections ($CV_{real} = 0.014$, $CV_{shuffled} = 0.009$), as is summation efficacy ($CV_{real} = 0.041$, $CV_{shuffled} = 0.037$; 1000 shuffles). Because each ORN axon makes synapses at many locations on the dendritic tree, the differences among an ORN's many synapses tend to average out. This makes the qualitative properties of each connection quite similar (i.e., the properties that do not depend on synapse number).

In short, the major source of variation among ORN→PN connections is simply the number of synapses per connection. Our models indicate that this source of variation should produce a relatively large range in uEPSP amplitudes (*Figure 5H* and *Figure 3B*). Different ORNs in the same antenna should have quite different effects on the PN membrane potential, with the strongest ORNs outweighing the weakest ORNs by almost 10-fold. Thus, it seems that PNs do not assign equal weight to ORNs from the same antenna. Rather, the PN response is likely to be dominated by only a fraction of the ORN population.

## Connection noise in olfactory receptor neuron projections

In principle, the variation in ORN→PN connections might be the result of developmental noise. Alternatively, it might represent a strategy to optimize PN signals by weighting the most reliable ORNs more heavily. We cannot know for certain because we cannot compare the spike trains of the ORNs in question. However, we can find a clue by comparing different connections made by the same ORN. Paired electrophysiological recordings from sister PNs show that ORN spikes virtually always produce synchronous synaptic events in all ipsi- and contralateral target PNs (*Kazama and Wilson, 2009*). In essence, all sister PNs experience the same ORN spike trains, even if they are located on opposite sides of the brain. Therefore, if PNs weight ORNs according to their reliability, then those weights should be correlated across all five PNs in our sample. Our analyses thus far have indicated that synapse number is the main correlate of connection strength variability (*Figure 4* and *Figure 5*), so we will focus on synapse number here.

Correlations across PNs are easiest to assess if we first normalize each ORN's contribution by the total contribution from the antenna in question. For example, there are 26 ORNs in the right antenna, which collectively make 893 synapses onto PN1 on the right side. A typical ORN from the right antenna contributes 34 synapses, or 3.81% (34/893) of the total contribution from that antenna. By focusing on these percentages, we can make a fair comparison between all connections, because this normalization procedure controls for systematic right-left differences as well as ipsi-contra differences in synapse number.

Using this metric of connection strength (i.e., normalized synapse counts), we found positive correlations between the three PNs on the left side (*Figure 6B*), and also positive correlations between the two PNs on the right side (*Figure 6C*). Specifically, 7 of the 8 ipsilateral comparisons produced a significant correlation (Pearson's $r$ ranged from 0.44 to 0.78, p<0.05; the exception is left PN2 and left PN3, where $r$ = 0.36, p=0.07). However, we found no correlation between right and left PNs (*Figure 6B–D*; for the 12 contralateral comparisons $r$ ranged from −0.21 to 0.34, p>0.09)

The failure to find correlations among all PNs argues that ORN→PN connection strengths are not optimized to match some feature of each ORN's spiking behavior. If they were optimized, they would be correlated among all five PNs, because all PNs witness identical ORN spike trains (*Gaudry et al., 2013*). In other words, if some ORNs are more reliable than other ORNs in the same antenna, then all five PNs should assign greater relative weight to these inputs. Because we do not

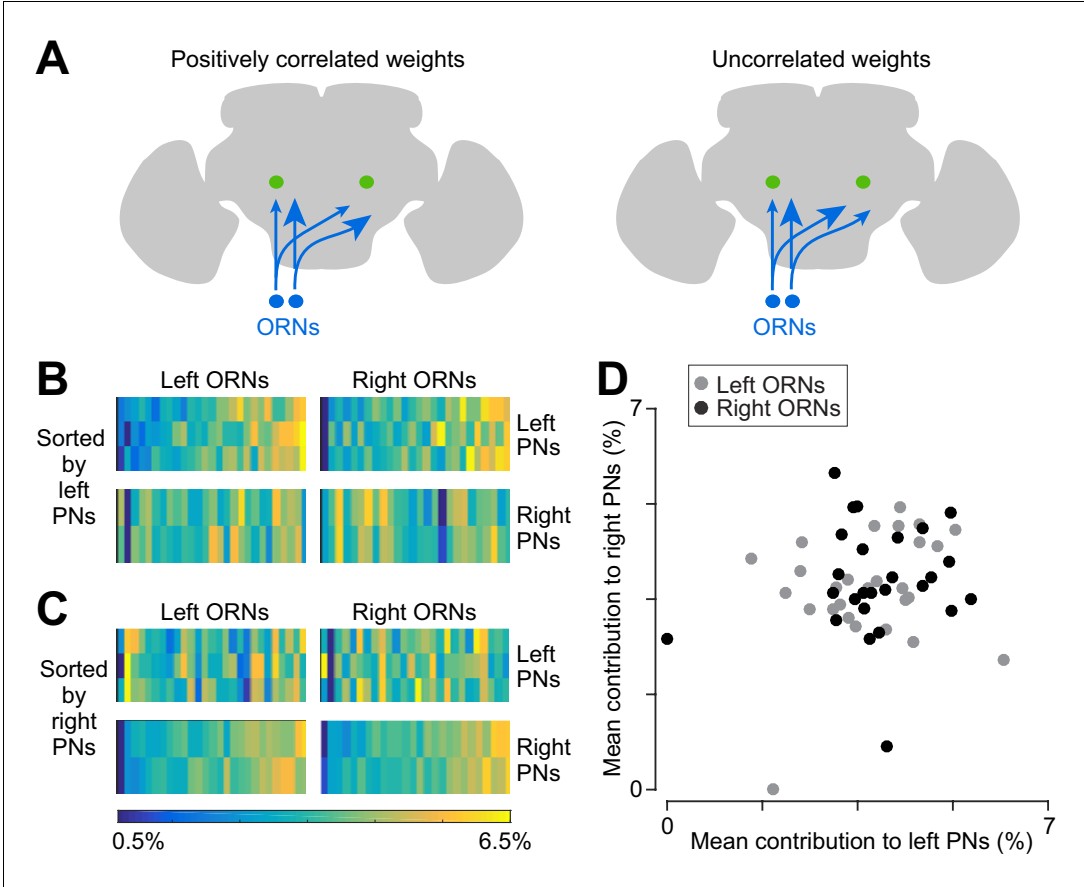

**Figure 6.** Correlated and independent variation in ORN wiring. (A) Schematics illustrating alternative scenarios: ORN connection weights may be correlated across PNs or uncorrelated. Arrowhead size represents the strength of ORN→PN connections. ORN spikes faithfully invade both ipsi- and contralateral axonal arbors, and so if connection weights are optimized to reflect the spiking properties of each ORN, then connection weights should be correlated across all ipsi- and contralateral PNs. (B) Contributions of individual ORNs to each PN's pool of ORN synapses. Values are expressed as the percentage contribution of each ORN to the pool of synapses from that antenna. Within each of these 10 vectors, values sum to 100%. Within an antenna, ORNs are sorted according to the average strength of all the connections that they form onto left PNs. Note that left PNs are correlated with each other, but not with right PNs. (C) Same data as in (B), but now sorted by average strength of connections onto right PNs rather than left PNs. Note that right PNs are correlated with each other, but not with left PNs. When we examined pairs of PNs in the same quadrant, we found that 7 of 8 PN pairs were significantly correlated with each other (Pearson's r ranges from 0.44 to 0.78, p<0.05, n = 27 or 26 unitary connections for each PN for each test, P values are corrected for multiple comparisons, m = 8 tests). The one exception was that left PN2 and left PN3 were not significantly correlated (Pearson's r = 0.36, p=0.07 after multiple comparisons correction). When we tested pairs PNs on opposite sides of the midline (again testing separately for correlations among right ORNs and left ORNs), we found that none of the 12 PN pairs were significantly correlated (Pearson's r ranges from −0.21 to 0.34, P always >0.09, n = 27 or 26 unitary connections for each PN for each test; tests were not corrected for multiple comparisons, as none were significant). (D) Average contribution of each ORN to the PNs on the right side, plotted against the average contribution of the same ORN to the PNs on the left. Percentages are calculated as in (B) before averaging across all the PNs on the same side of the brain. There is no significant correlation (Pearson's r = 0.18, p=0.20, n = 53 unitary connections for each PN).

observe this sort of correlation, then it seems likely that at least some of the variation in ORN→PN connections is due to 'connection noise' – that is, imprecision in synapse number specification. This connection noise should limit the accuracy of a PNs estimate of the stimulus based on incoming ORN spike trains.

## Functional implications of connection noise

Our results imply that at least some of the variation in ORN→PN connection strengths is unrelated to the content of ORN spike trains. Variation that is unrelated to ORN spiking is expected to degrade the performance of the organism on olfactory tasks. To estimate how large this effect could be, we performed an experiment using our compartmental models. Specifically, we asked how accurately an observer can judge the number of ORN spikes fired during a particular time window, based on a model PN's time-averaged voltage response (*Figure 7A*). It is important to note that all ORNs fire spontaneously even in the absence of odor (*de Bruyne et al., 2001*). Thus, at the perceptual threshold for odor detection, the olfactory system must be able to detect an odor based on an increase in ORN spikes above the expected number of spontaneous spikes.

The perceptual threshold for odor detection is known to be in the regime of low ORN spike numbers (*Bell and Wilson, 2016*; *Gaudry et al., 2013*). We therefore first focused on small odor-evoked increases in ORN spiking. We chose a 200 ms window for counting ORN spikes because it can take roughly this amount of time for a fly to show a behavioral response to an odor (*Bhandawat et al., 2010*; *Budick and Dickinson, 2006*; *Gaudry et al., 2013*; *van Breugel and Dickinson, 2014*). During this counting window, in one antenna, the entire DM6 ORN population fires an average of 12 spontaneous spikes in the absence of an odor (see Materials and methods). We want to estimate how connection noise can affect the ability to detect an increase in ORN spiking above this baseline.

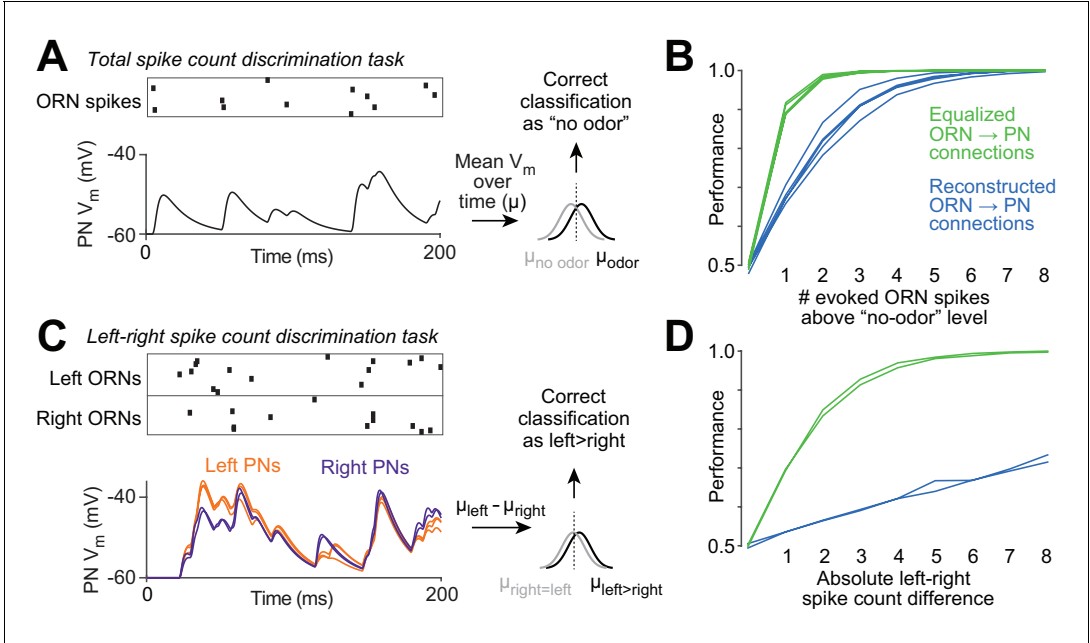

**Figure 7.** Functional consequences of variability in ORN wiring. (**A**) Schematic of 'odor detection' task. We measured how accurately a binary linear classifier could detect a small increase in ORN spike number, based on the time-averaged voltage in model PNs. Simulated Poisson spike trains were assigned to reconstructed ORN axons and fed into our PN models. Only ORNs ipsilateral to the PN were simulated. Over a 200 ms period, ORNs fired either 12 spikes (the average number of spikes during this time period that DM6 ORNs fire in the absence of an odor), or 13 spikes (representing a minimal odor stimulus). Based on the distribution of the time-averaged PN voltage (μ) in training trials, we classified each test trial as 'non-odor' or 'odor' (12 or 13 spikes). We repeated this many times for each of the five model PNs. The same procedure was used to measure accuracy when the 'odor' elicited increasing numbers of spikes (14 to 20 spikes). (**B**) Performance of the classifier as a function of the number of 'odor-evoked' spikes, above the baseline level of 12 spikes. Each blue line represents a different model PN. As in all previous simulations, the PN dendrite morphology and the locations of all ORN synapses are taken directly from our reconstructions. Green lines show that performance increases after we equalize the number of synapses per connection (by randomly reassigning synapses to ORN axons, so that all axons now have essentially equal numbers of synapses). (**C**) Schematic of 'odor lateralization' task. Here we simulated 'non-odor' activity (12 spikes) in one antenna and 'odor' activity (13–20 spikes) in the other antenna. Classification was based on the difference between PN voltage values on the left and right (cell-averaged μ_left − cell-averaged μ_right). (**D**) Performance of the classifier as a function of the number of 'odor-evoked' spikes (the left-right asymmetry). The two blue lines represent an odor stimulus in either the right antenna or the left antenna. Green lines show that performance increases after we equalize the number of synapses per connection.

We simulated ORNs as independent Poisson spike generators, each with the same average spike rate. Each ensemble ORN spike pattern was fed into a model PN, and we measured the time-averaged voltage response of the PN (*Figure 7A*). Even when ORN spike numbers are held constant, the time-averaged PN voltage responses vary from trial to trial, because the timing of ORN spikes varies. We repeated this procedure many times, and then trained a classifier to use the PN response to label trials as 'no odor' (12 ORN spikes per antenna) or 'odor' (13–20 spikes per antenna). We repeated this for all five PNs and their associated ipsilateral ORNs. Average performance was poor when the number of 'odor-evoked' spikes was small, and performance improved as the number of odor-evoked spikes was increased (*Figure 7B*, blue lines).

To estimate how much the natural variation in ORN→PN connection strengths degrades PN performance, we repeated this task, but now equalizing the number of synapses per ORN→PN connection. Specifically, we randomly reallocated ORN→PN synapses to presynaptic axons so that the number of synapses per axon was as equal as possible, but with each PN dendrite receiving synaptic input at all the same physical locations as before. Now performance was markedly improved (*Figure 7B*, green lines). This simulation shows that the normal variation in the structure of ORN→PN connections can impair the ability of PNs to transmit information about ORN spike counts.

We also considered an odor lateralization task (*Figure 7C*). This time, the ORN spike count in one antenna was held at the 'no odor' level (12 spikes), and ORN spike counts in the other antenna were either held at the same level (12 spikes) or else driven up by a lateralized 'odor' (to 13–20 spikes). In this simulation, each PN received input from all its presynaptic ORNs (both ipsi- and contralateral ORNs). Based on the difference between the mean voltage in right PNs versus left PNs, we trained a classifier to detect when spike number was different in the left and right ORN pools. Interestingly, performance in this task was especially poor (*Figure 7D*, blue lines) because the classifier operates on the difference between two variables (right and left), which are contaminated by independent connection noise. Taking the difference between these variables amplifies the effect of connection noise. Averaging the activity of PNs on the same side of the brain confers only a limited benefit because wiring noise is correlated among these PNs. When we artificially equalized the number of synapses per connection by randomly reassigning synapses to ORNs so that the number of synapses per axon was as equal as possible, we found that performance increased dramatically, as expected (*Figure 7D*, green lines). This simulation shows that the normal imprecision in the structure of ORN→PN connections can impair the ability of PNs to transmit information about right-left differences in ORN spike counts, particularly because connection noise is correlated within each half of the brain, but uncorrelated across the midline.

## Discussion

### Strengths and limitations of our data

An olfactory glomerulus represents a discrete neural network. The small spatial scale of this network allowed us to comprehensively reconstruct the connections between every excitatory principal cell (i. e., every ORN and PN). We also benefitted from existing *in vivo* electrophysiological measurements of these neurons. In particular, because we know the specific resistance and capacitance of the PN membrane (*Gouwens and Wilson, 2009*), as well as the amplitudes of uEPSPs and mEPSPs (*Kazama and Wilson, 2008*), we were able to construct highly-constrained compartmental models based on our EM reconstructions. These models served as analytical tools to examine the functional impact of structural variations among synaptic connections. Conclusions arising from these models were robust to measured variations in the model parameters derived from electrophysiological experiments (see Materials and methods).

The models we consider in this study have several limitations. One limitation is that our models include only ORN→PN synapses. ORNs contribute almost 75% of the synaptic input to PNs, with the remaining 25% arising mainly from local neurons (*Figure 1F*). Local neuron synapses could change the integrative properties of PN dendrites, but this would be difficult to capture in a model at present, even if we had reconstructed all the local neurons in the circuit, because local neurons have diverse and complicated spiking patterns (*Nagel and Wilson, 2016*), and it seems likely that only a subset of local neurons provide most of the input to PN dendrites (*Berck et al., 2016*).

A second limitation we faced was our lack of knowledge about the active properties of PN dendrites, if any. In principle, voltage-gated conductances in PN dendrites might alter the integrative properties of PNs. However, active properties are unlikely to play an especially large role in PN synaptic integration, because the current-voltage relationships in PNs are fairly linear (*Gouwens and Wilson, 2009*), whereas in other *Drosophila* neurons these relationships can be strongly nonlinear (A.W. Azevedo and R.I. Wilson, unpublished observations). Thus, our passive models are good first approximations in this case.

Yet another limitation is the inability to directly measure the conductance at individual synapses. Our models assume that conductance is identical at every ORN→PN synapse. This is a reasonable assumption, given that our candidate anatomical proxies for synaptic conductance (T-bar volume and postsynaptic contact area) turned out to be relatively uniform across connections (i.e., the CV of these measurements across connections was relatively low; *Figure 1—figure supplement 6D*). In the future, it will be interesting to investigate the anatomical correlates of synaptic conductance in more detail.

A final limitation is that PN dendrites may be distorted by sample preparation. EM fixation and preparation tends to reduce neuropil volume, but this effect is likely isotropic throughout the sample and mainly diminishes extracellular space (*Van Harreveld and Khattab, 1969*; *Korogod et al., 2015*). It is therefore likely that our estimated dendritic diameters are good first approximations.

## Structural correlates of connection strength

We can think of connection strength as being determined by three major factors. The first is the conductance at each synapse within the connection. The second is the total number of synapses in the connection. The third is the filtering of those conductances by the postsynaptic dendrite. We cannot directly measure the conductance at each synapse, but our anatomical measurements suggest that synapse-averaged conductance is relatively uniform across connections (*Figure 1—figure supplement 6*). Our reconstruction provides direct information about the other two factors (synapse number and dendritic filtering), and our results imply that the first of these is particularly important.

Notably, we found that the number of synapses per connection was strongly correlated with the strength of ORN→PN connections. The comparison between ipsi- and contralateral synapses represents the clearest example of this correlation, because we know from prior electrophysiology experiments that ipsilateral connections are 30–40% stronger (*Gaudry et al., 2013*). Here, we discovered that ipsilateral connections contain 35% more synapses per connection (*Figure 4C*). This result argues that the difference in the number of synapses per connection is the main difference between ipsi and contra connections. By extension, we can infer that there is not a sizeable ipsi-contra difference in synaptic conductance. We also found no ipsi-contra difference in the way that synapses are filtered by the PN dendritic tree, as evidenced by the fact that there is no ipsi-contra difference in simulated mEPSP amplitudes (*Figure 4E*). In short, the number of synapses per connection is the dominant mechanism underlying the systematic functional difference between ipsi- and contralateral connections.

Interestingly, from the perspective of an individual PN, there was little variation across ORN connections in the average strength of the synapses that comprised each connection (*Figure 5E*). The efficacy of mEPSP summation at the level of uEPSPs was also notably consistent across these connections (*Figure 5G*). Thus, insofar as our models accurately represent the structure of each PN dendrite, it predicts that dendritic filtering has an essentially uniform effect on all ORN→PN connections. This uniformity arises because each connection is composed of many synapses, and synapses made by a given ORN axon tend to be placed onto the dendrite in a relatively unbiased fashion. In essence, each connection is composed of many quasi-random 'samples' of dendritic filtering properties, and so the average effect of dendritic filtering is similar across connections.

## Co-variation in cell number, dendrite size, and synapse number

Although *Drosophila* neural networks are sometimes regarded as highly stereotyped, the number of neurons in such a network is actually variable. In the optic lobe, a recent EM study analyzed 7 repetitions of a modular neural network that normally contains 23 uniquely identifiable cells. In 3 of the 7 networks, one cell that ought to be present was in fact missing (*Takemura et al., 2015*). In the case of one missing cell, a homologous cell in a neighboring column sent an extra branch into the vacated

space, where it received synapses from the normal presynaptic partners of the missing cell. Thus, when a cell is missing, there can be compensatory changes in wiring.

Variations in antennal lobe PN numbers have been inferred previously based on Gal4 expression patterns (*Grabe et al., 2016*; *Tanaka et al., 2004*). Indeed, based on Gal4 expression, we find that there can be two, three, or four PNs in glomerulus DM6, with the 'typical' situation (three PNs) occurring only 67% of the time (*Figure 1—figure supplement 2*). The brain selected for large-scale serial section EM turned out to contain three PNs on the left side and two PNs on the right side. In the glomerulus with fewer PNs, we found that PN dendrites were larger, perhaps because they had more space to fill. Moreover, there was also a compensatory increase in synapse numbers per PN, so that the total number of synapses per glomerulus was similar on the left and right.

Our compartmental models allowed us to infer the functional consequences of these concerted changes in PN dendrite morphology and ORN→PN synapse numbers. Remarkably, in the glomerulus with only two PNs, the up-regulation in synapse number was neatly balanced by the increased size of PN dendrites. As illustrated by classic work at the neuromuscular junction, increasing the size of a postsynaptic compartment produces a lower input resistance, and so each quantum of neurotransmitter produces a smaller depolarization (*Katz and Thesleff, 1957*). In each of the larger PNs, a presynaptic ORN spike should release more quanta than normal, but the postsynaptic voltage response to each quantum will be smaller. As a result, each PN has the same average response to an ORN spike.

Counterbalanced effects like this can result from homeostatic mechanisms. For example, in the larval ventral nerve cord, a decrease in presynaptic neurotransmitter release can elicit compensatory growth in postsynaptic dendrites (*Tripodi et al., 2008*). Of course, we cannot be sure that this is a case of homeostasis; perhaps there is just a fixed allocation of ORN synapses per glomerulus, and this is why the number of these synapses is equal on the left and the right (*Figure 2B*). However, there is direct evidence that the electrical properties of the dendrites of antennal lobe PNs can instruct changes in ORN→PN connections. A previous study used cell-specific $K^+$ channel overexpression to decrease a PN's input resistance, and found a compensatory increase in unitary excitatory synaptic currents at ORN connections onto that PN (*Kazama and Wilson, 2008*). That result demonstrated that PN dendrites can up-regulate synaptic currents to compensate for reduced dendritic excitability.

Together, these findings suggest the following scenario. When one PN failed to develop, the remaining PNs grew larger dendrites, and then synapse number increased to compensate for increased dendrite size. This scenario is consistent with the well-described instructive role of PN dendrites in ORN axon development: PN dendrites form a glomerular map prior to the arrival of migrating ORN axon terminals (*Jefferis et al., 2004*).

This scenario is reminiscent of the 'size matching' principle that governs the development of vertebrate neuromuscular junctions, where the size of a muscle is matched to the size of the axon's terminal arborization, thereby ensuring that large muscles (with low input resistance) receive a larger quantal content per presynaptic spike (*Kuno et al., 1971*; *Lichtman et al., 1987*). At the developing vertebrate neuromuscular junction, the expansion of the postsynaptic cell seems to be primary, with the elaboration of the presynaptic arbor occurring in response (*Balice-Gordon and Lichtman, 1990*).

## Developmental noise in wiring

We found that the number of synapses per ORN→PN connection was quite variable. As a result, ORN→PN connections gave rise to simulated uEPSP amplitudes ranging from 1.6 to 10 mV (*Figure 5H*). If all ORNs were functionally identical, this sort of variation would be non-optimal, because each PN's response would be dominated by only a fraction of its presynaptic ORN axons. Indeed, our simulations showed that this sort of variation can substantially impair a PN's ability to accurately transmit information about total ORN spike counts, as well as right-left differences in ORN spiking (*Figure 7*). Our simulations suggest that connection noise may be a factor limiting perceptual acuity.

The discovery of variability per se is not surprising: it was already clear that connections in the *Drosophila* brain can be variable. In the medulla of the optic lobe, the CV of synapse number per connection ranges from 0.08 to 0.87 (computed across all connections of a given type; *Takemura et al., 2015*). In principle, variation across connections of a given type may be taken as evidence of developmental noise (imprecision), or else evidence of adaptive plasticity (precision).

For example, systematic variations in upstream input (inherited from earlier layers of visual processing) might drive adaptive activity-dependent changes in the number of synapses per connection in the optic lobe. Here, we focused on primary afferent synapses rather than synapses deep in the circuit, so any systematic upstream variations are limited to variations in ORN spike trains. Moreover, we know that all DM6 PNs witness identical ORN spike trains, because ORN spikes travel faithfully across the midline to invade both ipsi- and contralateral glomeruli (*Kazama and Wilson, 2009*). This fact allowed us to test the hypothesis of adaptive plasticity by reconstructing all the synapses that each ORN axon made onto all PNs. We found that the variation in synapse numbers was not faithfully correlated across all PNs (*Figure 6*), and so some of this variation is likely random – that is, unrelated to ORN activity, and caused by a fundamental imprecision in the processes that specify the number of synapses per connection.

Intriguingly, we found that synapse number variations were correlated across sister PNs on the same side of the brain, even though they were uncorrelated on the opposite sides of the brain. In principle, this might be evidence of incomplete adaptive plasticity – plasticity that works at ipsilateral connections but somehow fails at contralateral connections. More likely is the scenario of correlated developmental noise – e.g., some ORN axons may simply arrive sooner at the ipsilateral glomerulus, and so may form more physical contact with ipsilateral PNs, and thus more synapses. This sort of correlated developmental noise may be one reason why sister PNs on the same side of the brain display such high levels of correlated electrical noise (*Kazama and Wilson, 2009*). As we show here, sister PNs on the same side of the brain are dominated by the same pool of ORNs. These ipsilateral sister PNs converge onto higher-order neurons, which are especially sensitive to correlations in sister PN spike times. As a result, sister PN spike timing correlations represent a functionally-relevant constraint on circuit function which can affect both the speed and accuracy of odor stimulus responses (*Jeanne and Wilson, 2015*).

Previous studies have highlighted other examples of seemingly non-optimal neural wiring patterns. These have generally been examples of individual neurons following highly tortuous paths (*Lu et al., 2009*; *Otopalik et al., 2017*). Tortuosity is energetically costly (*Chklovskii et al., 2002*), and it is difficult to see any functional benefit to tortuosity in these cases, suggesting that it may simply be the result of an imprecise developmental process. Our study extends this idea by providing evidence of imprecision at the level of synaptic connectivity, not just imprecision in the path that a neuron takes to find a synaptic target.

## Comparison between adult and juvenile networks

In the larval antennal lobe, there are 21 glomeruli, as compared to ~50 in the adult. Moreover, each glomerulus in the larva is relatively simple: it contains just one ORN axon and one uniglomerular PN dendrite. The larval antennal lobe connectome has just been reconstructed (*Berck et al., 2016*), and it is instructive to note the differences between the adult and the larval versions of the same circuit.

One difference concerns the structure of individual synaptic connections. In the larva, the average ORN→PN connection contains ~70 synapses. whereas in the adult, it contains 23 synapses. Thus, the increased number of ORNs in the adult is partly compensated by a decrease in the number of synapses per connection.

Another difference is in the control of ORN output. In the larva, almost all synapses onto ORNs arise from multiglomerular neurons. In the adult, most synapses onto ORNs arise from multiglomerular neurons, but a substantial minority arise from PNs and ORNs. This suggests that the adult network may exert more complicated control of ORN neurotransmitter release.

## Variations in the connectome: signal and noise

The brain's computational power would be substantially reduced if all synaptic connections were identical. From this perspective, systematic variations in synaptic connections are evidence of the brain's functional capacity – the capacity to match a connection's strength to its required function. In this study, we had an unusual opportunity to discover systematic variations in a particular connection type, because there are many ORN→PN connections per brain (260 connections in the glomeruli we reconstructed). By studying many instances of the same connection type, we were able to discover several systematic variations. Our results indicate that systematic variations in connection strength arise largely as a result of differences in the number of synapses per connection. We show how

systematic ipsi-contra differences can enable odor lateralization, while systematic correlations between synapse number and dendrite size can equalize the response of different PNs to an average ORN spike.

On the other hand, unsystematic wiring variations ('connection noise') must limit the capacity of every neural system. Some of this imprecision can be balanced by homeostatic changes to other parameters, including synaptic parameters (*Prinz et al., 2004*; *Roffman et al., 2012*). Our findings provide insight into the mechanisms underlying such compensatory changes, but our results also argue for the existence of residual non-optimal wiring variations that can impair neural computations.

Large-scale EM offers an unprecedented opportunity to study all these variations – and co-variations – in neural network wiring. *Drosophila melanogaster* is likely to be the next organism whose brain is fully mapped at the connectomic level. As such, it provides an opportunity to gain insight into the causes and consequences of systematic and noisy variations in network architecture.

## Materials and methods

### EM material preparation

The brain of an adult *Drosophila melanogaster* female (aged 8–10 days post-eclosion, genotype GH146-GAL4/+; UAS-CD2::HRP/+) was immobilized by cooling on ice, and then submerged in a drop of fixative (2% paraformaldehyde/2.5% glutaraldehyde in 0.1 M cacodylate buffer with 0.04% $CaCl_2$ for membrane stabilization). The head capsule was opened to allow fixative to access the brain before dissection (*Meinertzhagen, 1996*). Following dissection, the brain was processed for serial section transmission EM. The dissected brain was post-fixed and stained with 1% osmium tetroxide/ 1.5% potassium ferrocyanide, followed by 1% uranyl acetate, then lead aspartate (*Walton, 1979*), then dehydrated with a graded ethanol series, and embedded in resin (TAAB 812 Epon, Canemco). A total of 1917 serial thin (<50 nm) sections were cut on an ultramicrotome (Leica UC7) using a 35 degree diamond knife (EMS-Diatome) and manually collected on 1 × 2 mm dot-notch slot grids (Synaptek) that were coated with a pale gold Pioloform support film (Ted Pella), then carbon coated and glow-discharged. Grids were subsequently post-stained with uranyl acetate (saturated) in 50% methanol and 0.2% lead citrate.

### Large-scale TEM imaging and alignment

We imaged the 1917 sections using a custom-built Transmission Electron Microscope Camera Array (TEMCA) (*Bock et al., 2011*; *Lee et al., 2016*). Acquired at 4 nm/pixel in plane, this amounted to 50 terabytes of raw data comprising 250 million cubic microns of brain and >4 million (4000 × 2672 pixel) camera images. Magnification at the scope was 2000×, accelerating potential was 120 kV, and beam current was ~90 microamperes through a tungsten filament. Images suitable for circuit reconstruction were acquired at a net rate of 5–8 MPix/s.

The series was aligned using open source software developed at Pittsburgh Supercomputing Center (AlignTK) (*Bock et al., 2011*; *Lee et al., 2016*). AlignTK uses linear normalization to enhance image contrast. With individual eight bits/pixel image frames, a histogram of pixel values was generated for each 64 × 64 pixel subregion in a frame. 'Black' and 'white' levels were chosen every 64 pixels in *x* and *y* at roughly the 0.5% and 99.5% levels in the cumulative distribution function for the surrounding 256 × 256 region. These black and white levels were adjusted and smoothed on overlapping frames using a relaxation method to remove intensity discontinuities between frames. Each voxel in the final aligned volume was bilinearly interpolated from four adjacent pixels in a chosen raw frame and then linearly normalized to a 0–255 range using black and white values bilinearly interpolated from the coarse grid of adjusted black and white levels. No blending was used between overlapping frames or between sections.

The aligned series was then imported into CATMAID (*Saalfeld et al., 2009*) for distributed online visualization and segmentation. Within the portion of the volume spanned by the DM6 glomeruli, there were nine single-section losses, 2 instances of 2-section losses, 1 instance of 3-section losses, and 2 instances of 4-section losses (losses refer to consecutive sections). Across the entire series of 1917 sections there were 50 single-section losses, 16 instances 2-section losses, 10 instances of 3-section losses, 3 instances of 4-section losses, 3 instances of 5-section losses, and one instance of a

6-section loss. One reason why we chose the DM6 glomerulus for reconstruction is that there were no 5- or 6-section losses intersecting this glomerulus. Folds, staining artifacts, and cracks occasionally occurred during section processing, but were typically isolated to edges of our large sections and therefore did not usually interfere with manual segmentation.

## Synapse size analysis (T-bar volume and postsynaptic contact area)

We randomly selected 5 ORNs from the left antenna and 5 ORNs from the right antenna. Each of these 10 ORNs formed synaptic connections with all 5 DM6 PNs ($n_{total\ synapses}$ = 683, $n_{total\ postsynaptic\ pn\ profiles}$=1106). Volumes of interest around each synapse were exported from the CATMAID environment (512 pix $\times$ 512 pix $\times$ 51 sections or 2.05 $\times$ 2.05 $\times$ 2.04 µm volumes centered on CATMAID connectors marking each synapse). We then used itk-SNAP (http://www.itksnap.org/) to segment all pixels corresponding to T-bars and postsynaptic membranes. PN postsynaptic membranes were marked and area calculated depending on the orientation of the synapse relative to the plane of sectioning. If the section was cut perpendicular to the synaptic cleft, PN membranes were marked with a line. The length of this line in pixels was then multiplied by pixel resolution (4 nm) and average section thickness (40 nm) to give an area in $nm^2$. If, however, a section was cut obliquely or parallel to the synaptic cleft, the PN membrane was segmented as a polygon and the number of pixels in the polygon was multiplied by pixel area (16 $nm^2$). *En face* or obliquely cut synapses were identified by serial sections that starkly transitioned from a presynaptic specialization hosting a vesicle pool, to a distinctly different postsynaptic cell, typically with an intervening section of electron dense area representing the postsynaptic density and/or synaptic cleft. Because synapse orientation can change across sections, the decision about how to mark and measure PN membrane areas was made on a section-by-section basis. In cases where a single PN was postsynaptic to the same ORN T-bar multiple times at a single synapse (~10% of synapses), we used the same label for all profiles and subsequently manually sorted the segmentations corresponding to different profiles. All pixels corresponding to T-bars were marked with a polygon, and the number of pixels attributed to a T-bar was then multiplied by pixel area and section thickness (16 $nm^2$ $\times$ 40 nm) to yield its volume. Four annotators made the measurements of T-bar volume and postsynaptic contact area, with three annotators independently measuring each T-bar and postsynaptic contact. The annotator whose measurements were overall closest to the average was selected as the benchmark annotator, and all the measurements of the other three annotators were weighted to counteract their overall bias relative to the benchmark annotator (with biases inferred from their independent measurements of the same synapses). Finally, the three bias-corrected measurements of each T-bar and each PN postsynaptic membrane area were averaged together.

## Reconstruction and verification

We reconstructed ORNs and PNs in the EM data set by using CATMAID to manually place a series of marker points down the midline of each process to generate wire-frame models of axonal and dendritic arbors (*Lee et al., 2016*; *Saalfeld et al., 2009*). We identified synapses using a combination of ultrastructural criteria – namely, the existence of a presynaptic T-bar, presynaptic vesicles, a synaptic cleft, and postsynaptic densities. The presence of most, though not necessarily all, of these features over multiple sections was required for a synapse to be annotated. PNs were easily identifiable based on having dendrites restricted to DM6 and axons projecting into the inner antennocerebral tract. Annotators first identified putative PNs within the DM6 volume based on their large-gauge processes. Annotators then traced each arbor to completion, or until a violation of the above-mentioned PN identification criteria made it clear the cell was not a DM6 PN. While tracing, annotators also comprehensively marked all input and output synapses the arbor participated in. This procedure follows a previously described and validated protocol for manual tracing in serial section TEM data sets such as ours (*Schneider-Mizell et al., 2016*). Subsequently, all profiles presynaptic to each PN were annotated in the same manner. Profiles were categorized as ORNs if they arrived in the antennal lobe from the antennal nerve bundle; all such profiles innervated a single glomerulus either bilaterally (in the case of 51 ORNs) or unilaterally (in the case of 2 ORNs). Because ORN axons travel from the antennal nerve to DM6 in a distinctive trajectory, it was clear even before an ORN exited DM6 that it was indeed an ORN. Profiles not annotated as either PNs or ORNs were categorized as multiglomerular neurons; all such profiles presynaptic to PNs were traced until they crossed

the border of DM6 before their tracing was suspended. The multiglomerular neurons in DM6 likely include local neurons, multiglomerular PNs, and neurons innervating the antennal lobe from other brain regions (extrinsic neurons). For all reconstructed neurons included in our analysis, at least one additional independent annotator(s) verified the tracing by working backward from the most distal end of every process. Because ORN axons traveling through the anterior commissure run parallel to the plane of sectioning and are relatively fine caliber, tracing errors were more likely in this structure than other regions of the antennal lobe. To verify our reconstructions in this region, we re-imaged sections containing DM6 ORN axons in the commissure at higher magnification (20,000×). We then aligned the higher-resolution data set to the lower-resolution data set, and an additional independent annotator (who was blind to the original reconstructions) was assigned to trace all the DM6 axons through the commissure.

## Labeling DM6 PNs for cell counting

To count the number of PNs that are normally present in a DM6 glomerulus, we labeled and counted these cells in five additional flies. We used *NP3481-Gal4* to drive expression of photoactivatable GFP (PA-GFP) (*Datta et al., 2008*; *Patterson and Lippincott-Schwartz, 2002*) in DM6 PNs, along with several other PN types (*Tanaka et al., 2012*). By selectively photoactivating within the neuropil of DM6, we could photolabel the DM6 PNs alone, and then count the number of PNs innervating that glomerulus. Photoactivation was performed using a custom built two-photon laser-scanning microscope. We first imaged the antennal lobe at 925 nm and low laser power to identify DM6. After defining volumes of interest restricted to the core of the glomerular neuropil based on these images, PA-GFP was photoconverted by imaging through the volume with 710 nm light. In each photoactivation block, we moved through the z depth of the volume of interest with 0.25 μm steps. Each glomerulus was subject to three photoactivation blocks at intervals of five to ten minutes. After PA-GFP is photoactivated in the glomerular neuropil (i.e., axons and dendrites), it diffuses into the somata of cells. We then imaged each brain using an Andor XD spinning disk confocal microscope equipped with a Yokogawa CSU-X1 spinning disk unit and a Zyla 4.2 CMOS camera. For this imaging we used laser illumination at 488 nm. We found 3.1 ± 0.57 (mean ± SD) photoactivated DM6 PN somata on each side of the brain in five flies (10 glomeruli total).

## Compartmental models

To generate compartmental models, we first inflated our wire-frame reconstructions of PNs to create a representation of the entire volume of each dendritic segment. We did this in the CATMAID environment by estimating the average neurite caliber between each pair of skeleton branch points. We then used this value as the radius of the distal branch point (relative to the soma) and all nodes leading up to the proximal branch point. Dendritic segments between branch points were therefore represented as cylinders of uniform radius matched to the observed caliber of the neurite. This process was repeated until the entire skeleton was inflated.

Next, we exported PN morphologies along with ORN input synapse locations as neuroML 1.8.1 models (*Gleeson et al., 2010*). Using the software tool neuroConstruct (*Gleeson et al., 2007*), we defined PN membrane properties and synaptic conductances, and subsequently exported these models to the NEURON simulation environment (*Hines and Carnevale, 1997*). Additionally, we used neuroConstruct to remesh the PN models to ensure each segment had an electrotonic length between 0.1 and 0.0001. Each model PN was comprised of 1814–3148 cables, each with a measured diameter taken from the data (these are termed 'sections' in the NEURON modeling environment); these cables were then further subdivided into 14,510–23,502 compartments for numerical simulation purposes, so that each cable was comprised of multiple compartments having equal length and diameter (termed 'segments' in the NEURON modeling environment). We gave PN membranes uniform, passive properties. A previous experimental study (*Gouwens and Wilson, 2009*) used electrophysiological recordings from PNs to derive values for their specific membrane resistance (20.8 kΩ cm$^2$), specific membrane capacitance (0.8 μf/cm$^2$), and specific axial resistivity (266.1 Ω cm). The measurements of that study were taken from DM1 PNs, so in order to use these measurements in our study, we assume that the intensive (size-independent) properties of DM6 PNs are similar to those of DM1 PNs. The ORN to PN synaptic conductance waveform we used was also adapted from this study. This was modeled as the sum of two exponentials, with a rising time constant of 0.2 ms

and a decay time constant of 1.1 ms. *Gouwens and Wilson (2009)* modeled the maximum synaptic conductance as having a value of 0.27 nS (it was incorrectly reported in that paper as $2.7 \times 10^{-4}$ nS, but the value used was actually 0.27 nS); we used a value of 0.1 nS here because this produced more realistic uEPSP amplitudes (on average roughly 5 mV; *Kazama and Wilson, 2008*).

To measure the amplitude of individual mEPSPs, we sequentially activated individual synapses in the modeled PN dendrite, allowing enough time (200 ms) between events for the PN membrane potential to decay to baseline. We recorded PN voltage responses either at the cell body (somatic mEPSPs) or within the compartment where the synaptic conductance was activated (dendritic mEPSPs). To simulate unitary EPSPs, we synchronously activated all synapses from an individual ORN. In this case we always measured the PN voltage response at the cell body (somatic uEPSPs).

It is worth asking whether these models are robust to measured variations in the model parameters derived from electrophysiological experiments. *Gouwens and Wilson (2009)*, measured the specific intracellular resistivity, membrane resistance, and membrane capacitance of three antennal lobe PNs ('Cell 1', 'Cell 2', and 'Cell 3'). There were variations in these three cells, but overall Cell three was intermediate, so we initially used values fit from Cell three to construct the five model PNs that we used throughout this study. As a check, we constructed five alternate model PNs using values fit from Cell 1, and also five more alternate model PNs using values fit from Cell 2. When we used these alternate PNs, we obtained results that were overall the same as the results we obtained with the PNs fit to Cell 3 values. The only difference is that we found a very small but statistically significance left-right difference in uEPSP amplitudes with the models that were fit to Cell 2 values, whereas we did not find any statistically significant left-right difference in uEPSP amplitudes with the models fit to Cell 1 or Cell 3 values ($P_{Cell\ 1} > 0.07$, $P_{Cell\ 2} > 0.02$, $P_{Cell\ 3} > 0.7$, permutation test, $n = 156$ left and 104 right unitary connections). Other results were unchanged. Namely, the systematic ipsi-contra difference in uEPSP amplitudes was similar in all cases ($P_{Cell\ 1} = 0.007$, $P_{Cell\ 2} = 0.006$, $P_{Cell\ 3} = 0.0059$, paired-sample *t*-test, $n = 5$ PNs). Also, correlations between synapse number and mean uEPSP amplitudes were similar in all cases (Pearson's *r* ranged from 0.984 to 0.997 (Cell 1), 0.998 to 0.999 (Cell 2), 0.993 to 0.999 (Cell 3). *P*-values ranged from $1.59 \times 10^{-39}$ to $2.94 \times 10^{-56}$ (Cell 1), $3.85 \times 10^{-60}$ to $1.22 \times 10^{-72}$ (Cell 2), $9.78 \times 10^{-48}$ to $1.34 \times 10^{-65}$ (Cell 3) after Bonferroni-Holm correction for multiple comparisons, $m = 5$ tests).

## Classifier performing an 'odor detection' task

We modeled the ORNs ipsilateral to each PN as independent Poisson spike generators. Spike trains in which two spikes in the same neuron occurred at an interval <4 ms were rejected, in order to simulate a refractory period. Each modeled ORN spike train was randomly assigned to an ORN axon. The mean spontaneous firing rate of the DM6 ORNs cannot easily be measured from single-sensillum recordings on the antenna because the sensillum which houses DM6 ORNs is the ab10 sensillum (*Couto et al., 2005*), which is small and has remained undetected in surveys of single-sensillum physiology (*de Bruyne et al., 2001*; *Hallem and Carlson, 2004*). We therefore measured the mean spontaneous firing rate of DM6 ORNs by measuring the rate of large spontaneous EPSCs in voltage-clamp recordings from DM6 PNs in a re-analysis of data collected from DM6 PNs for a previously published study (*Gaudry et al., 2013*). In all recordings we analyzed, the contralateral antenna had been removed just before the recording, so PNs were receiving spiking input from ipsilateral ORNs only. The rate of these events was 58 Hz, and so dividing by 26.5 ORNs, we obtain 2.2 spikes/s/ORN. This method (using spontaneous EPSC rates) has been shown to provide a good agreement with single-sensillum recordings of ORN spike rates in the case of a different glomerulus, glomerulus DM4 (*Kazama and Wilson, 2008*). Multiplying the estimated spontaneous ORN spike rate of 2.2 spikes/s by a window of 200 ms and rounding to the nearest integer yields a basal 'no odor' value of 12 spikes.

In each trial, the ipsilateral population of 26 or 27 ORNs fired a specified number of spikes within the first 200 ms, ranging from 12 to 20, and no spikes in the second 200 ms. This allowed the PN membrane potential to return to baseline by the end of each trial. We then used the time-averaged PN membrane potential, recorded at the soma, to train and test a linear classifier to label trials as 'odor' (12 ORN spikes) or 'no odor' (13–20 ORN spikes). For each of the five model PNs, we ran 2500 trials for training and 2500 independent trials for testing. We initially performed this exercise using model PNs based directly on our reconstructions, where each ORN spike activated all synapses attributed to that ORN in our reconstruction. Subsequently, we artificially equalized the number of

synapses per ORN axon in the following manner. We defined a pool of synapse locations on the PN dendrite corresponding to all the synapses made by ipsilateral ORNs onto that PN. We then allocated synapses arbitrarily and evenly to ORNs by drawing from this pool (without replacement) until synapses were re-allocated. When the total number of ipsilateral ORN synapses was not evenly divisible by the number of ipsilateral ORNs, the remainders were randomly assigned to simulated ORNs. When synapse numbers are artificially equalized in this manner, summation efficacy increases, and so uEPSP amplitudes increase. In order to keep average uEPSP amplitudes the same as before, the maximum conductance of our synapse models were reduced to 0.0958 nS. Thus, our synapse equalization procedure made uEPSP amplitudes more uniform across connections without changing the mean uEPSP amplitude. As before, for each of the five model PNs, we ran 2500 trials for training and 2500 independent trials for testing. Synapses were independently re-allocated in each trial.

### Classifier mimicking an 'odor lateralization'' task

The procedure here was the same as for the spike count classifier, except in every trial we modeled ORNs in the right antenna as well as ORNs in the left antenna, and all PNs received both right and left ORN input. On any given trial, the ORNs in one antenna fired 12 spikes, and the ORNs in the other antenna fired 12 to 20 spikes. In every trial, both right and left ORN spike trains were fed into the dendrites of all 5 PN models. We then used the means of the resulting PN responses to train and test a linear classifier to detect trials where there was a right-left asymmetry in ORN spike count. In the equalized case, we equalized synapse numbers independently for each PN-antenna combination. For example, we forced all ORNs in the right antenna to have equal contributions to right PN1, and we also forced all ORNs in the left antenna to have equal contributions to right PN1, but the average asymmetry between ipsi- and contralateral connections was preserved. In the equalized case, the maximum conductance of our synapse models were reduced to 0.0958 nS, as before.

### Statistics

The number of observations of any given variable was dictated by the number of cells and synapses in glomerulus DM6, and so was not predetermined using statistical methods. Statistical comparisons between sample distributions were done with Permutation tests (i.e. Monte Carlo-based Randomization tests) unless otherwise noted. Permutation test were used because they do not assume the underlying distributions are normal, and because observations do not need to be independent. For Permutation tests, we computed the incidence of differences between means or Pearson's linear correlation coefficient of randomly drawn samples from combined sample distributions exceeding the empirical difference.

### Code and data availability

The aligned EM data set and model files can be accessed as resources at: https://neurodata.io/data/tobin17. Custom code is available at: https://github.com/htem/tobin17 (*Tobin, 2017*); a copy is archived at https://github.com/elifesciences-publications/tobin17.

## Acknowledgements

We thank S Bellou, R Caplan, C Dekker, K Hern, M Johnson, P Starkey, Z Tweed for primary tracing and reconstruction; W Copeland, Y Hasanat, J Kiperman, B Sanders for targeted synapse segmentation; E Raviola for discussions and advice; M Reed and C Bolger for technical support at the beginning of the study; A Cardona and S Saalfeld for making the CATMAID project openly available; and T Kazimiers for assistance in optimizing our CATMAID instance. We also thank B Graham and R Torres for programming, A Pandya for help with alignment, and G Hood for alignment pipeline support. JT Vogelstein, A Baden, E Perlman and R Burns helped make the data freely available. We thank S Druckmann, M Pecot, A Samuel, and members of RIW lab for feedback on the manuscript. This work was supported by NIH grant R03 DC013622 (to WCL), NIH grant R01 DC008174 (to RIW), a Harvard Brain Initiative Collaborative Seed Grant (to RIW and A Samuel), the Bertarelli Program in Translational Neuroscience and Neuroengineering, the Edward R and Anne G Lefler Center, and the Stanley and Theodora Feldberg Fund. RIW is an HHMI Investigator. The project is solely the responsibility of the authors and does not necessarily represent the official views of the NIH.

## Additional information

### Funding

| Funder | Grant reference number | Author |
|---|---|---|
| National Institutes of Health | R03 DC013622 | Wei-Chung Allen Lee |
| Harvard Brain Initiative Collaborative Seed Grant | | Rachel I Wilson |
| Bertarelli Program in Translational Neuroscience and Neuroengineering | | Wei-Chung Allen Lee |
| Edward R. and Anne G. Lefler Center | | Wei-Chung Allen Lee |
| Theodora Feldberg Fund | | Wei-Chung Allen Lee |
| Howard Hughes Medical Institute | | Rachel I Wilson |
| National Institutes of Health | R01 DC008174 | Rachel I Wilson |

The funders had no role in study design, data collection and interpretation, or the decision to submit the work for publication.

### Author contributions

WFT, Conceptualization, Data curation, Software, Formal analysis, Validation, Investigation, Visualization, Methodology, Writing—original draft, Writing—review and editing; RIW, Conceptualization, Resources, Supervision, Funding acquisition, Writing—original draft, Project administration, Writing—review and editing; W-CAL, Conceptualization, Resources, Data curation, Software, Formal analysis, Supervision, Funding acquisition, Validation, Investigation, Visualization, Methodology, Writing—original draft, Project administration, Writing—review and editing

### Author ORCIDs

William F Tobin, http://orcid.org/0000-0001-7417-049X
Wei-Chung Allen Lee, http://orcid.org/0000-0002-4618-295X

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
