## [Decision Letter]

Thank you for submitting your article "Wiring variations that enable and constrain neural computation in a sensory microcircuit" for consideration by *eLife*. Your article has been favorably evaluated by Eve Marder (Senior Editor) and three reviewers, one of whom, Liqun Luo (Reviewer #1), is a member of our Board of Reviewing Editors. The following individual involved in review of your submission has agreed to reveal their identity: Albert Cardona (Reviewer #2).

The reviewers have discussed the reviews with one another and the Reviewing Editor has drafted this decision to help you prepare a revised submission.

We append the full reviews from all three reviewers below. We hope that you can address each of the critiques in your revision. We would like to clarify one item regarding "synapse size" (the first critique of reviewer 3). Upon further discussion among reviewers, reviewer 3 has the following additional comments: "The relationship between synapse size, as well as synapse number, and connection strength is part of a general background from *Drosophila* studies. […] Not only would this address an important component of pathway strength, it would acknowledge an important issue that might not be obvious to readers who work on different nervous systems."

*Reviewer #1:*

This is an excellent paper integrating new EM reconstruction data with modeling and previous electrophysiology studies to produce general insight about the relationship between the structure, physiological properties, and functions of microcircuits. The authors focus on reconstructing olfactory receptor neurons (ORNs) and projection neurons (PNs) that innervate one of the fifty pairs of glomeruli in the *Drosophila* antennal lobe. Besides generating valuable quantitative ultra-structural data about the synaptic connectivity, the authors produced a compartmental model that simulates how ORN inputs are integrated in PNs, and used it to analyze three interesting aspects from their data: how PNs compensate for the difference in their numbers (particularly interesting), why ipsi-lateral ORNs exhibit stronger connections with PNs than do contra-lateral ORNs, and what is the functional implication for variations in synapse numbers between individual ORN→PN pairs. The authors have also explicitly discussed limitations of their study (e.g., their model does not take into account the inhibitory input or variations in strengths of individual synapses). Among the recent surge of EM reconstruction studies, this one ranks as one of the best to integrate structure and physiological properties of a well-studied model circuit, and as such should be of broad interest to the neuroscience community.

*Reviewer #2:*

Tobin et al. tackle the timely question of the nature of variation in synaptic connectivity and how neural computations are affected by variations in the synaptic structure of a circuit.

The manuscript reports on the relationship between neuronal function and dense anatomical reconstructions as obtained from serial section electron microscopy. Among the many interesting results, some of them are likely to make of this paper an instant classic. For example, the authors quite clearly demonstrate that the strength of a unitary connection strongly correlates with the number of synapses from an axon to a dendrite. The entire and rather large emerging field of connectomics in *Drosophila* will look up to this paper for data on how to translate anatomical synaptic contacts to functional connectivity strength.

In addition, the authors exploit the opportunity to address the neural circuit basis of odor lateralization, that is, the ability of the animal to distinguish odor intensities between the left and right antennae. The bilateral nature of the olfactory receptor neuron (ORN) axons could a priori make this task impossible, but the authors discovered a systematic variation in that the number of synapses is larger in the ipsilateral rather than the contralateral ORN-PN connection, which the authors show is sufficient to enable computing whether the odor concentration is higher on either side of the animal.

Importantly, this study touches on how neurons of the same cell type interact with neurons of a second cell type, a connectivity arrangement important in large nervous systems such as that of vertebrates. The authors show data in support of an inherent, underlying flexibility in the dimensions of the dendrites of the postsynaptic neurons relative to the number of postsynaptic neurons, given a similar number of presynaptic neurons.

The statement, quote: "we find that right and left PNs have uniform average voltage responses to ORN spikes, in spite of their marked differences in dendrite size and synapse number." might turn out to be of extraordinary value, given that it suggests the details of an underlying developmental program that can cope with changes in the number of neurons, allowing the population to explore the fitness landscape towards the best number as appropriate given environmental conditions.

In summary, this work is largely complete and self-contained, and a pleasure to read.

*Reviewer #3:*

References are to text pages, not pdf.

Wiring variations that enable and constrain neural computation in a sensory microcircuit.

First, I take my hat off to the authors for their devoted study of AL circuits using such labor-intensive methods. For this submission, anatomy is the bedrock, while the modeling aspects are more conjectural. In particular, I would submit that the weakest link is the main story line, the link between synaptic structure and synaptic strength. Even so, it ought to be possible to relate all parameters of the connectome to those of the modeling, provided we have all the requisite measures. To this end, the authors provide important data on dendritic caliber, which has generally not been treated extensively in the literature on *Drosophila* neurons. While it's easy to count synapse number and plot distributions, however, so far missing from the authors' story is synapse size. Does this vary in any consistent way between (pre)synaptic sites, and do the postsynaptic densities, which I presume are also visible, co-vary with the size of presynaptic contact? The size of the dendrite may co-vary with these, but is not itself a proxy for synapse size. If the authors are to claim a relationship between synapses number and the strength of synaptic transmission, I think these additional parameters should be included, and hope they can be mined with relative ease from existing image data, even if I imagine these are all fixed modules.

Introduction, fourth paragraph. Additional important discussion on wiring accuracy in brains is to be found in Takemura et al. (PNAS 112 (Rall, 1964): 13711-13716, 2015) which the authors cite but only later in support of missing cells in medulla columns. The PNAS paper includes reference to comparisons between left/right homologues in *C. elegans* and comparisons between synapse numbers in neighboring medulla columns in *Drosophila* both of which bear on the authors' data.

The authors' conclusions concerning systematic and unsystematic variations in wiring (Introduction, last paragraph and later) stand in contrast to those obtained from the fly's visual system (Takemura et al., 2015). In fact, the medulla analyses indicate the high order of synaptic and wiring accuracy of which *Drosophila* is also capable in constructing the synaptic circuits in its brain, in that case in the visual system. Accuracy, or invariance, in the case of the latter is presumably driven by the need to transmit reliably at the high temporal frequencies demanded by vision, a requirement that can be less demanding in the olfactory pathway.

Feedforward. I find "feedforward" in the following construction unnecessary and confusing: "along with all the feedforward excitatory synapses onto these cells." (Introduction, fourth paragraph). 'Feedforward' and 'onto' sound to me to be contradictory because feedforward could refer to the synaptic outputs of the cell in question. I think what the authors want to say is "along with all the excitatory inputs (synapses) onto these cells".

"Three experts independently identified glomerulus DM6 in the EM data set by visual inspection based on published light-level maps". The account is remarkable more for what it does not say than what it does say. I find the documentation of how the authors selected DM6 neurons for reconstruction to be insufficient. Although confocal data clearly reveal the antennal lobe's glomerular map, based especially on synaptic and genetic labelling, such data are deceptive for EM studies, which reveal all profiles not their selective features. In particular there are no clear glial boundaries between glomeruli, and glomeruli, which the authors fail to mention, and can only be identified unambiguously with reference to the map of surrounding glomeruli, or possibly from Cartesian coordinates, and I think the authors should acknowledge this. Quite possibly the authors have gained sufficient expertise from physiological recordings to identify specific glomeruli but this is in the intact living antennal lobe. Glomeruli also partly overlap each other in the depth of the lobe neuropile. Moreover, we are not told why DM6 itself was chosen over other glomeruli, and as a rationale the authors may want to cite their own physiological studies. These questions are central to the authors' analysis and should be made clear in the text, because while I am sure that great care was taken, and we can be reassured that three experts made the same identification, in fact it's not a trivial problem at all at EM level. Most readers would not know or understand this because confocal maps are so convincing, because most readers never see EM, and because most may erroneously assume that there is a clear glial boundary surrounding each glomerulus. It's also quite possible to target DM6 in a living prep, which studies from the Wilson lab have done repeatedly, but the contours of the antennal lobe's surface also differ from the 2D images from EM sections. These issues of 'why' and 'how' need to be clearly addressed. The text of the Methods should enable another worker to repeat the authors' observations. In particular, how can the authors provide the reader the clear assurance that they have actually targeted the same glomerulus on left and right sides of the brain, and that the left/right asymmetries they find are not the outcome of choosing different glomeruli on the two sides. I fully accept, by contrast, that the authors scrupulously identify in the EM dataset ORNs and PNs from their respective arbors and axon trajectories.

Figure 1. Related: although valuable as an EM overview, Figure 1 provides insufficient information to identify DM6, especially in the z axis. Was the entire glomerulus reconstructed and how did the authors decide where its boundaries lay?

Orphans: To what does the rate of orphans refer? The number of independent fragments that were presynaptic to PNs but could not be connected to an identified cell? How many synapses did these contribute? Were any orphaned at both ends, having no contact with either identified ORN or PN? If so, what proportion?

The claim (subsection “Cells comprising the glomerular micro-circuit”, last paragraph) that false continuations between fragments are easily detected and corrected during independent review is more likely to be an attribute of the arbor than of the reconstruction strategy, since larval neurons in *Drosophila* tend to have relatively more stout neurites with distal extremities that bear most synapses. Thus, the conclusion that they are therefore very rare in the final reconstruction may be less warranted than is claimed.

Presynaptic T-bars. Please comment on the range of structural phenotypes exhibited by presynaptic T-bars in Figure 1—figure supplement 3, or from the authors' analysis of synapses in serial sections.

Table 1. Subsection “Relative abundance of connection types”, first paragraph. I do not find Table 1, which is needed to interpret Figure 1—figure supplement 4.

Synaptic numbers, subsection “The structure and strength of excitatory connections”. Documentation of synapse numbers is good, and interpretation is generally appropriate, but the authors should review their text carefully to check the tendency to equate synapse number with synapse strength, and to distinguish semantic reference to these two strictly. It's also important to remember the processing depth introduced by feedback synapses, for which transmission gain reflects not only the number of feedforward synapses but also upstream synapses. Thus, in addition to the gain of individual synapses, feedback circuits inherit the signal gain from their synaptic inputs.

Adjacency matrix in Figure 1—figure supplement 4 resembles a number of such matrices that have now been published, with the top numbers around 50 synapses, in line with those in other brain regions, which the authors could indicate. Each intercept represents the site of presumed transmission between a single pre- and postsynaptic neuron, but the text should acknowledge that because each presynaptic site has several postsynaptic elements, a number of intercepts in the matrix necessarily are coordinately linked. In fact, I find no easy metric for synaptic divergence revealed in this figure, or elsewhere in the text. How many dyads, triads etc. are incorporated in the matrix. Alternatively, how many monads?

Related to the preceding point: Figure 1—figure supplement 3. These are single images of synapses that fail to reveal profiles above and below the plane of section. Yet clearly there are multiple dendrites at single release sites, as can indeed be seen in some sections. This information is given in Figure 4—figure supplement 1 but only as averages that compare Ipsi and Contra ORNs.

The authors state that "All synaptic conductance events had the same size and shape." This presumably assumes that those conductance events were shared at the same presynaptic site? Do the authors imagine that each postsynaptic dendrite sees the same cloud of neurotransmitter release, or is the cloud shared amongst the several dendrites. If that number varies, does this mean that transmission varies with the number of dendrites, or is the dendritic signal unaltered; how does the outcome affect the authors' modeling? I don't think there is an easy answer, but these issues should at least be acknowledged, and as far as possible discussed.

Figure 1—figure supplement 4. In addition, I find that using black for 0 synapses is confusing, not least because I find it hard to discriminate differences in tone for dark intercepts having few synapses, and recommend replotting this matrix using white for intercepts having 0 synapses. Alternatively, reverse the entire spectrum of tones, so that 50+ is black and 0 is white.

Methods. Of the 1917 sections cut, we are told, in the subsection “Large-scale TEM imaging and alignment”, how many sections were lost, apparently 145, from 1 to 6, about 7% of the total series – so roughly equal to the number of orphans, but not how many sections were needed to encompass the DM6 glomeruli, and thus not therefore what proportion of those were drops. Losing 6 sections (0.3µm) is very damaging to the continuity of any series, but at least the authors are honest in reporting them. How many of these drops accounted for the occurrence of orphans, one wonders, and were these equal on left and right DN6 sides? The authors should I think try to address these points.

---

## [Author Response]

*We append the full reviews from all three reviewers below. We hope that you can address each of the critiques in your revision. We would like to clarify one item regarding "synapse size" (the first critique of reviewer 3). Upon further discussion among reviewers, reviewer 3 has the following additional comments: "The relationship between synapse size, as well as synapse number, and connection strength is part of a general background from Drosophila studies. It happens that in flies from many studies now that most synapses are of uniform size, so that pathway strength is indeed approximated by synapse numbers without reference to their sizes. Even so, synapse size is rarely reported and compared between neuropile regions. To demonstrate modularity of synapse size it would only be necessary to measure ~40 synaptic profiles in fair cross section and 20 in longitudinal section, in order to determine variation in synapse contact sizes, and to compare with other published reports. Not only would this address an important component of pathway strength, it would acknowledge an important issue that might not be obvious to readers who work on different nervous systems."*

We agree that synapse size is an important issue. There are two questions here that we can address with our dataset. First, how much does synapse size vary? Second, how do synapse size variations correlate with other anatomical variations?

To address these questions, we randomly selected 10 ORNs (5 from the left and 5 from the right antenna), and manually segmented all of their synapses onto all 5 of our reconstructed PNs (n_T-bars_= 683, n_pn profiles_= 1106). At each synapse, two features were measured, the ORN presynaptic T-bar volume and the PN postsynaptic contact area (PSCA) for all postsynaptic PN profiles present. To increase the precision of these measurements, each T-bar volume and PSCA was independently measured by three different annotators, and the measurements of different annotators were then combined after weighting each annotator’s measurements according to their overall measurement bias across the whole data set (relative to the other annotators).

The results of our synapse size analyses are now shown in a new supplement (Figure 1—figure supplement 6). Overall, we found that the two measures of synapse size (T-bar volume and postsynaptic contact area) were significantly (though weakly) correlated. They were also significantly correlated with the number of postsynaptic profiles per T-bar. This means that there must be mechanisms that produce coordinated variations in all three variables on a synapse-by-synapse basis (T-bar volume, postsynaptic contact area, and number of postsynaptic profiles).

That said, when we compared different ORN→PN connections (by averaging the size of all the synapses belonging to the same connection), we found that synapse size was only a minor source of variation among connections (Figure 1—figure supplement 6). The main source of variation among connections was still the number of synapses per connection (Figure 5). In other words, what distinguishes one connection from another is primarily the number of synapses it contains, and not the size of those synapses.

*Reviewer #1:*

*This is an excellent paper integrating new EM reconstruction data with modeling and previous electrophysiology studies to produce general insight about the relationship between the structure, physiological properties, and functions of microcircuits. The authors focus on reconstructing olfactory receptor neurons (ORNs) and projection neurons (PNs) that innervate one of the fifty pairs of glomeruli in the Drosophila antennal lobe. Besides generating valuable quantitative ultra-structural data about the synaptic connectivity, the authors produced a compartmental model that simulates how ORN inputs are integrated in PNs, and used it to analyze three interesting aspects from their data: how PNs compensate for the difference in their numbers (particularly interesting), why ipsi-lateral ORNs exhibit stronger connections with PNs than do contra-lateral ORNs, and what is the functional implication for variations in synapse numbers between individual ORN→PN pairs. The authors have also explicitly discussed limitations of their study (e.g., their model does not take into account the inhibitory input or variations in strengths of individual synapses). Among the recent surge of EM reconstruction studies, this one ranks as one of the best to integrate structure and physiological properties of a well-studied model circuit, and as such should be of broad interest to the neuroscience community.*

We are grateful for this positive feedback.

We note that we do now look explicitly at anatomical parameters that may be proxies for the “strengths of individual synapses”. To be precise, we describe variations in synapse size parameters (T-bar volume, postsynaptic contact area). We find that while these parameters vary substantially across synapses, they vary only modestly across different synaptic connections, as noted above.

*Reviewer #3:*

References are to text pages, not pdf.

Wiring variations that enable and constrain neural computation in a sensory microcircuit.

*First, I take my hat off to the authors for their devoted study of AL circuits using such labor-intensive methods. For this submission, anatomy is the bedrock, while the modeling aspects are more conjectural. In particular, I would submit that the weakest link is the main story line, the link between synaptic structure and synaptic strength. Even so, it ought to be possible to relate all parameters of the connectome to those of the modeling, provided we have all the requisite measures. To this end, the authors provide important data on dendritic caliber, which has generally not been treated extensively in the literature on Drosophila neurons. While it's easy to count synapse number and plot distributions, however, so far missing from the authors' story is synapse size. Does this vary in any consistent way between (pre)synaptic sites, and do the postsynaptic densities, which I presume are also visible, co-vary with the size of presynaptic contact? The size of the dendrite may co-vary with these, but is not itself a proxy for synapse size. If the authors are to claim a relationship between synapses number and the strength of synaptic transmission, I think these additional parameters should be included, and hope they can be mined with relative ease from existing image data, even if I imagine these are all fixed modules.*

As requested by the reviewer, we have now measured synapse size parameters. We provide an overview of our analysis and our findings at the beginning of this reply document, where we respond to the Editor’s Summary (which highlighted this particular request). Synapse size analyses are now included in Figure 1—figure supplement 6.

*Introduction, fourth paragraph. Additional important discussion on wiring accuracy in brains is to be found in Takemura et al. (PNAS 112 (Rall, 1964): 13711-13716, 2015) which the authors cite but only later in support of missing cells in medulla columns. The PNAS paper includes reference to comparisons between left/right homologues in C. elegans and comparisons between synapse numbers in neighboring medulla columns in Drosophila both of which bear on the authors' data.*

*The authors' conclusions concerning systematic and unsystematic variations in wiring (Introduction, last paragraph and later) stand in contrast to those obtained from the fly's visual system (Takemura et al., 2015). In fact, the medulla analyses indicate the high order of synaptic and wiring accuracy of which Drosophila is also capable in constructing the synaptic circuits in its brain, in that case in the visual system. Accuracy, or invariance, in the case of the latter is presumably driven by the need to transmit reliably at the high temporal frequencies demanded by vision, a requirement that can be less demanding in the olfactory pathway.*

We agree that Takemura et al. (2015) should be cited more prominently. We had devoted space to this paper in the Discussion, but not in the Introduction. We now cite this paper prominently in the Introduction as an example of variability in neural wiring.

The reviewer suggests that the demands of visual processing require more precision in neural architecture than the demands of olfactory processing do. We do not think the relative demands of visual and olfactory computations are entirely clear at this point, and we would not agree with the idea that olfactory processing is undemanding. Odors can fluctuate at high temporal frequencies in natural turbulent settings, with plume hit durations of <20 ms or less. These temporal features are useful for navigation. Olfactory transduction in *Drosophila* is fast enough to follow fluctuations up to 10 Hz. Thus, olfaction is not much slower than vision. Olfaction is also very high dimensional, because a large number of physiolochemical dimensions are needed to describe the differences between distinct chemicals. This is yet another reason why it might be adaptive for the olfactory system to have a high rate of information transmission at the periphery. It is therefore particularly interesting that the wiring olfactory system should be imprecise.

We do not think the level of variation we describe here is notably larger than the imprecision described by Takemura et al. (2015). Takemura et al. described two kinds of variation:

1) Binary connection variation (missing connections, or connections that are normally absent yet occasionally present). In the optic lobe, Takemura et al. found that the rate of binary variation is about 1%. At ORN-to-PN synapses, we find that the rate is about 2%. (One ORN on the left failed to make a connection with the 2 right PNs, and one ORN on the right failed to make a connection with the 3 left PNs. The rate of missing connections is therefore 2+3 divided by ((26 ORNs+27 ORNs) × 5 PNs), or 2%.) Therefore, binary connection variation is not notably different in the two data sets.

2) Analog connection variation (variation in the number of synapses per unitary connection, across all instances of a particular connection type). In the optic lobe, Takemura et al. found that the CV of synapse number per connection ranges from 0.08 to 0.87, depending on connection type. At ORN-to-PN synapses, we find that the CV is 0.31. Therefore, analog connection variation is not notably different in the two data sets.

Given the variability described by Takemura et al., it is unsurprising that we also find variability in our reconstruction. The contribution of our study is not simply to describe variability per se. Rather, the contribution of this study is to provide evidence for both systematic and unsystematic kinds of anatomical variability. We do not need to assume that all variability is “noise” (or “error”). We show that certain anatomical parameters co-vary in a systematic way that ought to be adaptive (given the framework of our model), whereas other parameters vary in an unsystematic way that ought to be maladaptive (again, given the framework of our model).

*Feedforward. I find "feedforward" in the following construction unnecessary and confusing: "along with all the feedforward excitatory synapses onto these cells." (Introduction, fourth paragraph). 'Feedforward' and 'onto' sound to me to be contradictory because feedforward could refer to the synaptic outputs of the cell in question. I think what the authors want to say is "along with all the excitatory inputs (synapses) onto these cells".*

We thank the reviewer for pointing this out. Our intention was just to distinguish ORN→PN synapses from other types of excitatory synapses which are not “feedforward” (e.g., PN→PN synapses, PN→ORN synapses). In the revised manuscript, have revised to this phrase to read as follows: “along with all the feedforward excitatory synapses that these cells receive”.

*"Three experts independently identified glomerulus DM6 in the EM data set by visual inspection based on published light-level maps". The account is remarkable more for what it does not say than what it does say. I find the documentation of how the authors selected DM6 neurons for reconstruction to be insufficient. Although confocal data clearly reveal the antennal lobe's glomerular map, based especially on synaptic and genetic labelling, such data are deceptive for EM studies, which reveal all profiles not their selective features. In particular there are no clear glial boundaries between glomeruli, and glomeruli, which the authors fail to mention, and can only be identified unambiguously with reference to the map of surrounding glomeruli, or possibly from Cartesian coordinates, and I think the authors should acknowledge this. Quite possibly the authors have gained sufficient expertise from physiological recordings to identify specific glomeruli but this is in the intact living antennal lobe. Glomeruli also partly overlap each other in the depth of the lobe neuropile. Moreover, we are not told why DM6 itself was chosen over other glomeruli, and as a rationale the authors may want to cite their own physiological studies. These questions are central to the authors' analysis and should be made clear in the text, because while I am sure that great care was taken, and we can be reassured that three experts made the same identification, in fact it's not a trivial problem at all at EM level. Most readers would not know or understand this because confocal maps are so convincing, because most readers never see EM, and because most may erroneously assume that there is a clear glial boundary surrounding each glomerulus. It's also quite possible to target DM6 in a living prep, which studies from the Wilson lab have done repeatedly, but the contours of the antennal lobe's surface also differ from the 2D images from EM sections. These issues of 'why' and 'how' need to be clearly addressed. The text of the Methods should enable another worker to repeat the authors' observations. In particular, how can the authors provide the reader the clear assurance that they have actually targeted the same glomerulus on left and right sides of the brain, and that the left/right asymmetries they find are not the outcome of choosing different glomeruli on the two sides. I fully accept, by contrast, that the authors scrupulously identify in the EM dataset ORNs and PNs from their respective arbors and axon trajectories.*

The borders of antennal lobe glomeruli are delineated by a specialized class of glia (termed “ensheathing glia”; Doherty et al. 2009 J Neurosci29:4768-81). We agree with the reviewer that these glial ensheathments may be more or less clear in ssEM material, depending on how a specimen was prepared and imaged, and also depending on the identity of the glomeruli in question. In our material, the borders of many glomeruli are quite clear at low resolutions (please see Video 1–Video 2). We chose to focus on glomerulus DM6, in part because it is a prominent glomerulus, with an easily identifiable position relative to other antennal lobe glomeruli. In published maps of the antennal lobe, DM6 has a particularly consistent location (Laissue et al. 1999 J Comp Neurol405, 543-552; Couto et al. 2005 Curr Biol15, 1535-1547; Grabe et al. 2015 J Comp Neurol523, 530-544). We also chose DM6 because there is a wealth of physiological information regarding this glomerulus. Finally, DM6 was located in a region of the EM series where there are relatively few missing sections (detailed in the original Materials and methods). We have now revised the Methods to explain more clearly our rationale for choosing DM6. The fact that we had correctly identified a pair of homologous glomeruli on the two sides of the brain was ultimately confirmed when the reconstruction was done: we found that DM6 ORN axons crossed the anterior commissure to target the other glomerulus we had identified as DM6.

*Figure 1. Related: although valuable as an EM overview, Figure 1 provides insufficient information to identify DM6, especially in the z axis. Was the entire glomerulus reconstructed and how did the authors decide where its boundaries lay?*

We thank the reviewer for the opportunity to clarify this point. Some of the borders of DM6 (namely, the anterior, medial, and dorsal borders) were clear based on visual inspection (please see Video 1–Video 2). The posterior, lateral, and ventral borders were less clear, and so were ultimately defined by the dendrites of the reconstructed PNs. All the dendrites of the PNs we reconstructed completed filled the same volume (which is typical of most PNs and most glomeruli). As described in the Methods, we reconstructed every single profile presynaptic to every PN dendrite (at least to the border of DM6), and so we have high confidence that we identified every DM6 ORN. The purpose of Figure 1 is to provide an overview of an EM section at the level of DM6, but the identification of DM6 was, of course, done with high-resolution 3D data. We expect that specialist readers who are interested in this issue will examine the 3D data set at neurodata.io.

*Orphans: To what does the rate of orphans refer? The number of independent fragments that were presynaptic to PNs but could not be connected to an identified cell? How many synapses did these contribute? Were any orphaned at both ends, having no contact with either identified ORN or PN? If so, what proportion?*

We define orphans as fragments of reconstructed neurites that could not be connected to any neuron. We have now revised the text to clarify this definition. The reviewer is correct that orphans were independent fragments that were presynaptic to PNs but could not be connected to an identified cell. With our reconstruction strategy (reconstruct all inputs to PNs) one end is always anchored at a PN input synapse so we do not generate fragments orphaned on both ends. Therefore, by construction, the number of synapses contributed by orphans to PNs is essentially the number of orphans fragments themselves.

*The claim (subsection “Cells comprising the glomerular micro-circuit”, last paragraph) that false continuations between fragments are easily detected and corrected during independent review is more likely to be an attribute of the arbor than of the reconstruction strategy, since larval neurons in Drosophila tend to have relatively more stout neurites with distal extremities that bear most synapses. Thus, the conclusion that they are therefore very rare in the final reconstruction may be less warranted than is claimed.*

Although false continuations are possible, our reconstruction strategy favors minimizing false continuations at the risk of false terminations to maximize confidence in the connections we reconstruct. Annotators stop tracing

if a continuation becomes ambiguous. Moreover, all reconstructed neurons were reviewed by multiple independent annotators.

*Presynaptic T-bars. Please comment on the range of structural phenotypes exhibited by presynaptic T-bars in Figure 1—figure supplement 3, or from the authors' analysis of synapses in serial sections.*

We have now included a quantitative analysis of T-bar volumes in the revised manuscript (Figure 1—figure supplement 6), as described above. We feel that a detailed description of T-bar shapes is beyond the scope of this study, but we are planning to investigate this issue in the future.

*Table 1. Subsection “Relative abundance of connection types”, first paragraph. I do not find Table 1, which is needed to interpret Figure 1—figure supplement 4.*

The reviewer might not have been able to locate Table 1 because it is provided as a separate, machine readable Comma Separated Value (CSV) supplementary file.

*Synaptic numbers, subsection “The structure and strength of excitatory connections”. Documentation of synapse numbers is good, and interpretation is generally appropriate, but the authors should review their text carefully to check the tendency to equate synapse number with synapse strength, and to distinguish semantic reference to these two strictly.*

We note explicitly in the Introduction that the strength of a synaptic connection can depend on many anatomical features. The number of synapses per connection is just one of those features. One of the goals of our study was to infer which feature(s) are actually the largest contributors to observed physiological variations in connection strength. We begin with the fact that ipsilateral ORN→PN connections are known to be 30 – 40% stronger than contralateral connections (Gaudry et al. 2013), and we ask what anatomical features might account for this physiological difference. We find that synapse number per connection is about 27% higher on the ipsilateral side (Figure 4), while there is no systematic difference in the dendritic properties of ipsilateral and contralateral synapses (as judged by mEPSP amplitudes, which reflect synapse location on the dendrite and the electrical filtering properties of the dendrite itself, Figure 4). This result argues that synapse number is the main contributor to connection strength variations. Moreover, we found no significant difference between ipsi- and contralateral PN postsynaptic contact area. Contralateral T-bars turn out to be 8% larger than ipsilateral T- bars, but this difference is relatively modest, and it is accounted for by the fact that T-bars grow with the number of postsynaptic elements, and contralateral ORN projections target more non-PN profiles than ipsilateral ORNs do (Figure 4—figure supplement 1). Finally, when we just focus on ipsilateral connections, we again find that the number of synapses per connection is again the most variable anatomical feature of synaptic connections (CV = 0.31, Figure 5), with comparatively less variation in the overall dendritic properties of a connection (CV_mEPSP amplitude_=0.014, Figure 5; _CVsummation efficacy_=0.041, Figure 5) or the mean synapse size for each connection (_CVT-bar volume_=0.13, CV_postsynaptic contact area_=0.13, Figure 1—figure supplement 6). Taken together, these results are evidence that synapse number is the main anatomical feature contributing to connection strength variations, although other anatomical features also make a contribution. This is an important result for the field, because it establishes synapse number as a reasonable proxy for connection strength – with the caveat that this feature will certainly not capture *all* the variation in connection strength.

In the context of the manuscript, we use the phrase “connection strength” only when we are intentionally talking about physiological connection strength – either in the context of published studies that have measured synaptic responses directly, or in the context of our compartmental model, which uses parameters measured in published electrophysiological studies to extrapolate from our anatomical measurements to physiological predictions. We make the point that synapse number is a reasonable proxy for connection strength, but we also make clear that it is merely a rough proxy, because synapse number is not the only anatomical feature that contributes to variation among synaptic connections.

*It's also important to remember the processing depth introduced by feedback synapses, for which transmission gain reflects not only the number of feedforward synapses but also upstream synapses. Thus, in addition to the gain of individual synapses, feedback circuits inherit the signal gain from their synaptic inputs.*

We agree with the reviewer that the feedforward connections we focus on in this study (ORN→PN connections) are not the only important elements of this circuit. We do report data on PN→PN synapses, ORN→ORN synapses, and PN→ORN synapses, although we do not focus on any of these connection types in the latter part of the study. We are also explicit about the fact that there are many cells in the circuit we have not fully reconstructed (cells we are calling multiglomerular cells, which are mainly local neurons). Certainly, some of these additional cells are likely to be mediating feedback. We anticipate that the feedback elements of this circuit (e.g., local neurons that mediate feedback inhibition) will be described in future ssEM studies.

*Adjacency matrix in Figure 1—figure supplement 4 resembles a number of such matrices that have now been published, with the top numbers around 50 synapses, in line with those in other brain regions, which the authors could indicate. Each intercept represents the site of presumed transmission between a single pre- and postsynaptic neuron, but the text should acknowledge that because each presynaptic site has several postsynaptic elements, a number of intercepts in the matrix necessarily are coordinately linked. In fact, I find no easy metric for synaptic divergence revealed in this figure, or elsewhere in the text. How many dyads, triads etc. are incorporated in the matrix. Alternatively, how many monads?*

We agree with the reviewer that this information should be reported. We have now added Figure 1—figure supplement 5 showing the distribution of polyady at ORN output synapses in DM6.

*Related to the preceding point: Figure 1—figure supplement 3. These are single images of synapses that fail to reveal profiles above and below the plane of section. Yet clearly there are multiple dendrites at single release sites, as can indeed be seen in some sections. This information is given in Figure 4—figure supplement 1 but only as averages that compare Ipsi and Contra ORNs.*

Please see the response to the reviewer’s previous point. We now make the existence of dyads, triads, etc. more explicit in Figure 1—figure supplement 5.

*The authors state that "All synaptic conductance events had the same size and shape." This presumably assumes that those conductance events were shared at the same presynaptic site? Do the authors imagine that each postsynaptic dendrite sees the same cloud of neurotransmitter release, or is the cloud shared amongst the several dendrites. If that number varies, does this mean that transmission varies with the number of dendrites, or is the dendritic signal unaltered; how does the outcome affect the authors' modeling? I don't think there is an easy answer, but these issues should at least be acknowledged, and as far as possible discussed.*

When we say, “All synaptic conductance events had the same size and shape”, we mean all conductance events, at all postsynaptic sites, period. In essence, our model assumes that the arrival of a spike at a T-bar always opens the same number of postsynaptic ion channels in all cells postsynaptic to that T-bar. We have revised the text of the manuscript to make this assumption as explicit as possible. This assumption is obviously a simplification, but model-building always involves some simplifications, and we think that this is not an unreasonable simplification. Certainly, it is relevant that some T-bars are apposed to a single postsynaptic profile, while others are apposed to two profiles (dyads) or three profiles (triads), or even more. However, we do not think that synaptic conductances are necessarily weaker in cases when more dendrites are “sharing” the same T-bar, because there is no reason why neurotransmitter concentrations are necessarily lower at dyadic or triadic connections, as compared to monadic connections. Notably, we find that T-bar volume grows with the number of profiles postsynaptic to the T-bar (Figure 1—figure supplement 6), which may indicate a mechanism whereby quantal content scales up to match the volume of the synaptic cleft.

*Figure 1—figure supplement 4. In addition, I find that using black for 0 synapses is confusing, not least because I find it hard to discriminate differences in tone for dark intercepts having few synapses, and recommend replotting this matrix using white for intercepts having 0 synapses. Alternatively, reverse the entire spectrum of tones, so that 50+ is black and 0 is white.*

We thank the reviewer for this suggestion. Please find Figure 1—figure supplement 4 updated in the revision.

*Methods. Of the 1917 sections cut, we are told, in the subsection “Large-scale TEM imaging and alignment”, how many sections were lost, apparently 145, from 1 to 6, about 7% of the total series – so roughly equal to the number of orphans, but not how many sections were needed to encompass the DM6 glomeruli, and thus not therefore what proportion of those were drops. Losing 6 sections (0.3µm) is very damaging to the continuity of any series, but at least the authors are honest in reporting them. How many of these drops accounted for the occurrence of orphans, one wonders, and were these equal on left and right DN6 sides? The authors should I think try to address these points.*

In the revised Methods we provide the following details regarding lost sections: “Within the portion of the volume spanned by the DM6 glomeruli, there were 9 single-section losses, 2 instances of 2-section losses, 1 instance of 3-section losses, and 2 instances of 4-section losses (losses refer to consecutive sections).” Certainly, lost sections are lamentable, but they are also not uncommon in this sort of data set. We chose DM6 specifically because *no* 5-section or 6-section losses intersected this glomerulus. Some of the orphan fragments our reconstruction are the consequence of section losses, but other orphan fragments are not adjacent to section losses and are instead due to ambiguities related to how the plane of a section intersects a neurite.